# The impact of sociocultural contexts on the knowledge, attitudes, and practices of adults living with HIV/AIDS in Ethiopia towards metabolic syndrome risks: A descriptive phenomenology study using the PEN-3 model

**Girma Tenkolu Bune**[1,2]*

1 School of Public Health (SPH), Dilla University(DU), Dilla, Ethiopia, 2 School of Public Health (SPH), Addis Ababa University (AAU), Addis Ababa (AA), Ethiopia

* girmatbune@gmail.com, girmatenkolu1973@gmail.com

## Abstract

### Introduction

The global HIV/AIDS initiative in Africa aims for eradication by 2030 and treatment for 95% of HIV-positive adults by 2025. Adult People living with HIV (PLWHs) face health complications, including metabolic syndrome (MS), which heightens the risk of non-communicable diseases (NCDs) and cardiovascular problems. WHO and UNAIDS advocate for the integration of NCDs into primary healthcare, yet addressing MS remains a significant challenge in Africa. The WHO's Global Action Plan aims to reduce chronic diseases by managing risk factors and promoting healthy lifestyles within this population. However, effectively promoting healthy lifestyles necessitates an understanding of the sociocultural contexts that influence behaviors related to MS. Therefore, this study investigates how sociocultural contexts influences on knowledge, attitudes, and practices of PLWHs in Ethiopia regarding MS prevention and associated lifestyle risks, utilizing the PEN-3 model as a sociocultural framework.

### Methods

The study utilized a deductive descriptive phenomenological approach, involving 32 voluntarily selected PLWHs who sought routine care at public health institutions from December 29, 2017, to January 22, 2018. Data collection was facilitated by experienced research and task teams using standardized guidelines for focus group discussions and in-depth interviews tailored to the research context. The collected text and survey data were managed with Atlas.ti and SPSS software and analyzed through thematic content analysis. Results were reported in accordance with the consolidated criteria for reporting qualitative research (COREQ) checklist.

### Results

A study of 32 HIV-positive adults found that knowledge, attitudes, and health-related behaviors were key factors in their health. Participants learned about metabolic syndrome (MS)

**Data Availability Statement:** The datasets used and/or analyzed during the current study presented

in this article and in the separately attached tables, figures and supplementary files.

**Funding:** The author(s) received no specific funding for this work.

**Competing interests:** NO authors have competing interests Enter: The authors have declared that no competing interests exist.

risks through mass media, peer discussions, and family education. However, there was a lack of awareness about the impact of HIV medications on MS and limited understanding of lifestyle factors for disease prevention. Attitudes reflect complex challenges for PLWHs in perceiving MS and its management. Health-related behaviors varied, with positive practices like fruit and vegetable consumption, regular exercise, and avoidance of harmful substances. Negative practices included sedentary lifestyles, raw meat consumption, alcohol, smoking, and 'Khat' use, which could negatively affect health outcomes. Addressing these culturally preferred behaviors is crucial for improving health among PLWHs.

## Conclusion

The study revealed a notable knowledge gap regarding metabolic syndrome (MS) and its risk factors, leading to inadequate health attitudes and practices. Sociocultural factors—such as beliefs, values, family dynamics, and community support—are crucial in shaping the knowledge, attitudes and practice of PLWHs toward the prevention and management of chronic diseases like MS. The finding suggested that addressing the sociocultural factors affecting HIV-positive individuals' knowledge and practices regarding metabolic syndrome requires a comprehensive, inclusive approach that emphasizes education, community involvement, policy reform, and a focus on reducing stigma.

## 1. Introduction

Global efforts to combat HIV/AIDS in Africa [1], strive to end the epidemic by 2030 and ensure timely treatment for 95% of HIV-positive adults by 2025 [2, 3]. However, elderly adult individuals living with HIV (PLWHs) may encounter obstacles stemming from age-related health issues like metabolic syndrome(MS) [4–6], which can increase the risk of non-communicable diseases(NCDs) [7] and heightened susceptibility to cardiovascular disorders(CVDs) [8–11]. The prevalence of these diseases is increasing globally, particularly in low- and middle-income countries [3–5, 12–15], impacting mortality rates in regions like Ethiopia where significant progress has been made in HIV care but NCDs remain a concern [16, 17].

The World Health Organization and UNAIDS support integrating chronic diseases (NCDs) into primary healthcare for people living with HIV (PLWHs) [18–20]. However, addressing risk factors like metabolic syndrome (MS) remains challenging in Africa due to limited NCD services, delivery gaps, lack of political commitment, coordination issues, funding shortages, inadequate healthcare worker expertise, and equipment deficits [16, 21–24]. The WHO's Global Action Plan highlights the need to reduce chronic diseases by tackling behavioral risk factors such as tobacco use, alcohol consumption, unhealthy diets, and physical inactivity [13, 16, 25]. It recommends investing in best practices in low- and lower-middle-income countries, as changing behaviors among HIV-positive individuals could significantly lower NCD-related mortality rates [16, 26, 27]. Behavior change programs should focus on individual actions while considering social and cultural contexts, which are essential for implementing evidence-based interventions to mitigate metabolic syndrome risks in PLWHs [21, 27–31]. Unfortunately, this aspect is often overlooked in resource-constrained settings [21, 32], particularly within research environments in Africa [21], hindering the long-term effectiveness of such interventions [30]. The PEN-3 model, established in 1989, offers a valuable framework for examining public health issues through a sociocultural perspective [21, 33].

The PEN-3 model, developed in African contexts, offers a framework for understanding cultural factors in public health [34]. It serves as a tool for analyzing health behaviors within

sociocultural settings, emphasizing the roles, values, and norms that promote or hinder health actions [21, 32]. This model underscores the need for culturally sensitive interventions and acknowledges the influence of family and community on behaviors [21, 32, 33]. The acronym 'PEN' represents three interconnected domains: the Cultural Identity Domain (CI), which focuses on individuals, families, and communities; the Relationships and Expectations Domain (RE), which highlights perceptions and influences; and the Cultural Empowerment Domain (CE), which considers both positive and negative aspects. These domains are essential for grasping health behaviors in sociocultural contexts, enabling a thorough examination of cultural dynamics and public health issues [35].

The PEN-3 model is a widely adopted framework for analyzing public health behaviors across various sociocultural contexts, applicable in areas such breast cancer screening [34], hypertension [21], medical education [33], and women's participation in cervical cancer screening [35]. Its utility has been demonstrated in diverse linguistic settings and geographical locations, notably in Africa, where it addresses cultural [21, 33–35]. Initiatives utilizing this model have included task-sharing for hypertension [21], management, understanding factors driving HIV self-testing among Nigerian youth [32], and advancing health communication regarding HIV/AIDS prevention in Tanzania [36]. Despite the model's broad application, a review of 166 research papers from Ethiopia indicates a significant gap in understanding cultural perspectives on health, particularly concerning HIV and non-communicable disease behaviors, even with the use of both quantitative and qualitative methodologies [37–40]. Most studies focused on metabolic syndrome among HIV patients predominantly employed quantitative observational methods, with few integrating qualitative approaches, such as phenomenology [3, 8, 38, 41, 42]. Notably, many studies failed to explore critical aspects, including knowledge, attitudes, and practices relating to lifestyle risks and sociocultural influences, often lacking appropriate theoretical frameworks [39, 40, 43–46]. While some research has attempted to utilize Western models [47], none have specifically aimed to analyze sociocultural contexts pertinent to the objectives of this study through the lens of the PEN-3 model.

Therefore, this study implements a qualitative descriptive phenomenological approach to explore the impact of sociocultural contexts on the Knowledge, Attitude, and Practices (KAP) of HIV-positive adults concerning lifestyle risks associated with metabolic syndrome and their long-term implications. It leverages the PEN-3 model and the KAP framework to comprehend how sociocultural factors influence health behaviors. The study aims to answer key research questions: How do PLWHs obtain information about metabolic syndrome and its associated lifestyle risks, and how does this knowledge influence their attitudes and behaviors? How do the acquired knowledge and formed attitudes towards metabolic syndrome and its risks affect lifestyle choices such as exercise, diet, physical activity, and other behaviors aimed at mitigating the risk of metabolic syndrome and its long-term consequences among PLWHs? For further clarity, please see study's conceptual framework attached as a supplementary (S1 Fig). The findings can help program developers and healthcare providers implement tailored behavioral modification strategies to reduce metabolic syndrome risks and improve health outcomes, particularly in Ethiopia and Sub-Saharan Africa. The study adhered to the SRQR completion checklist attached as supplementary information (S1 File) [48].

## 2. Methods

### 2.1. Research team and reflexivity

**2.1.1. Personal characteristics and roles.** Prior to commencing the study, three teams were established at the selected health institutions: a research team and two task teams. The research team consisted of the main author (Dr Girma Tenkolu, a lecturer and male researcher

at Dilla University) and two (male and female) public health professionals who were university staff members; all had MPH, 2nd Degrees and solid experiences in research and the profession. The main author moderated discussions while team members engaged in note-taking, audio-visual data documentation, and expanding field notes. The task teams, on the other hand, consisted of three health workers and two adherence counselors (one male and one female) from each ART clinic in the health facilities. Their role was to engage in the recruitment of subjects. Both teams were deployed in the research after taking of two days of on-site training.

**2.1.2. Relationship with participants.** Prior to the primary study starting, all participants were informed about the aims of the research and asked about their reasons for being interested in the topic of investigation. Since they've had some time to spend with the team members in the interim, there's a good chance that they will become close before the study even begins. Moreover, all staff members participated in the research after completing two days of on-site training.

**2.2. Study design and the theoretical framework.** The study adopted a deductive descriptive phenomenological qualitative approach, focusing on the epistemological and onto-logical aspects guiding the research inquiries. The case study technique was utilized within social epistemology contexts, highlighting the interconnections between individuals and their environment. A descriptive phenomenological study was conducted to gain a detailed under-standing of various inquiries such as the impact of personal knowledge on attitudes towards MS, strategies for mitigating its long-term effects, and how knowledge and attitudes influence dietary habits, physical exercise, smoking, and alcohol consumption practices. The study emphasizes how individual knowledge of MS, lifestyle risk factors, and long-term conse-quences can be shaped within the cultural identity domain, through the roles of individuals, extended families, and neighborhoods. Attitude development is influenced by perceptions of relationships and expectations concerning public health behavior, supported by personal, familial, and neighborhood elements. Practices aimed at managing or preventing MS can lead to positive, existential, or negative outcomes for individuals. By aligning research questions with the PEN-3 model, focusing on individuals, families, and neighborhoods helps uncover dietary patterns and other health-related behaviors shaped by upbringing. Understanding how diet and physical activity habits are influenced by individuals and their support systems, and the subsequent impact on health, offers insights into all three domains of the PEN-3 model. Descriptive phenomenology entails exploring and analyzing a specific phenomenon to under-stand subjective experiences of the participant PLWHs [49].

**2.2.1. Participant selection sampling.** All confirmed adult peoples living with the HIV (PLWHSs) in the Gedeo-zone were considered as a source population. Of which, those who fulfilled the inclusion and exclusion criteria presented below were selected to participated in the study. While HIV positive above 18 years old adult peoples and permanent resident of Gedeo-zone, who speak Amharic language and took the routine care at least for $>/=$ six months in the above referenced HIV clinics were used as inclusion criteria. Inactive partici-pants during the study time for various reasons and have restrictions that impeding their par-ticipation were correspondingly were used as exclusion criteria. To recruit the subjects, a purposive sampling method was used. Initially the subjects were informed orally about the study's aims before they took their routine care. Soon, the team appreciated the subjects' will-ingness' and immediately linked with the health workers to map out and enriched them with more information about the study. Subsequently, the securing of written consent and personal data collection process were quickly initiated and appointed for the discussion. The recruit-ment and data collection time schedules of the participant were undergone sequentially; run-ning first in Dilla University referral hospital (DURH), among the female group followed by the male, and Wonago health center (WHC), second. Note that, the interviewees were free to choose the time and location of their interviews.

**2.2.2. Method of approach.** All the above participants approached using face-to-face approaches with the help of the task teams while they visited the health care institution for their routine care.

**2.2.3. Sample size determination.** The sample size was determined based on saturation of idea, which was achieved as it gave similar answers for questions. As a result, we initially intended to organize six focus group discussions (FGDs) with six to eight male and female participants. However, due to idea saturation, we really only performed four FGDs and two impromptu in-depth interviews. Based on this, 32 persons in total were taken into consideration for the study out of the total anticipated number of participants.

## 2.3. Sampling methods and procedures

**2.3.1. Setting.** *2.3.1.1. Setting of data collection.* The study was conducted from December 29th, 2017, to January 22nd, 2018, in the Gedeo zone, located in the Southern Nations, Nationalities, and Peoples (SNNP) region; 360km to the south of Addis Ababa, the capital city of Ethiopia. The data collection was done in the purposefully chosen health institution: Dilla University referral hospital (DURH), and Wonago health center (WHC), which are located in the urban and rural area, respectively. To guarantee high-quality recordings and encourage candid communication, data collection took place in calm, safe, and peaceful environments.

Dilla Town, the administrative center, lies 369 km south of Addis Ababa, with a population density of 136 individuals per square kilometer and a 2.9% annual growth rate. Covering 134,700 hectares, the Gedeo Zone hosts an average population density of 853 individuals per square kilometer, totaling 1,148,517 people, with 82.9% residing in rural areas. Wonago Woreda stands out as one of the most densely populated Woredas, with 1,222 individuals per square kilometer.

The Gedeo people, an indigenous ethnic group in the region, maintain unique social and cultural practices, including their distinct Gedeo language closely linked to the Oromo language. Their societal structure revolves around clans with distinct traditions and esteemed elders. Their rich cultural heritage is evident in their traditional music, dance, crafts, and architectural style known as "Abaaye." Moreover, the Gedeo zone is home to a diverse population of ethnic groups practicing various cultures and languages such as Amaharic, Oromic, Sidamuu, Gurage, Silte, and others alongside Gedeoffa. While Gedeoffa is exclusive to the community, the predominant language for communication within the zone is the national and regional language, Amaharic.

Their religious beliefs blend traditional customs with Christianity and Islam, emphasizing reverence for ancestral spirits and the forces of nature. With a primary focus on agriculture, the Gedeo people grow coffee as their main cash crop while engaging in subsistence farming by cultivating crops like Enset (false banana), maize, and vegetables. Their traditional agroforestry system, encompassing 69.3% of the Gedeo Zone, integrates humans, trees, and various crops, with categories including Enset-tree-based agroforestry, Enset-coffee-based agroforestry, and coffee-fruit crops-tree-based agroforestry. Factors like population growth, migration, limited market access, and climate change require continual adaptation in agricultural practices in this region.

The study specifically targeted adults living with HIV (PLWHs) who sought care at public healthcare facilities in the Gedeo zone. These institutions were actively providing chronic care for HIV patients. As per the Gedeo zone ART Case Team Health Management Information System (HMIS) report, there were 141 health posts, 10 health centers, 1 referral and teaching hospital, and 3 primary hospitals in the zone, all government-owned. During the study period, 3,597 registered PLWHs in the area were receiving care from these institutions, with 629 being ART-naive and 2,968 already on ART [50].

*2.3.1.2. Presence of non-participants.* Nobody else than the participants and researchers was present throughout the discussion, and no one objected to becoming involved in the study either.

*2.3.1.3. Description of sample.* The place of residence and sex differences (being of consideration of male and female) participants was the important characteristics of the sample in this study.

## 2.4. Operational definitions

This study examines the sociocultural contexts that influence individual behaviors related to MS risk prevention and control of long-term complications. It focuses on understanding the knowledge, attitude, and practices (KAP) associated with MS risk prevention and control [51]. Knowledge is defined as adequate if one knows MS at all, at least three of its components, or risk factors and basic methods for prevention and control. Attitudes related behavior is favorable if knowledge leads to actions or behaviors intended to prevent or control either of the components of the syndrome. Conversely, unfavorable if no efforts are made to determine personal status or initiate practices to prevent or control MS. Practices related behavior is defined as adequate if one eats fresh fruits and vegetables > 3 days /week, exercises intentionally at least 3 days /week, and abstains from other bad lifestyle associated behaviors, such as smoking and alcohol use. Inadequate behavior is defined if fresh fruits and vegetables are eaten, and intentional physical exercise is fewer than 3 days /week, and unhealthy practices are engaged in. The study emphasizes the importance of understanding and promoting KAP in the context of MS risk prevention and control [52, 53].

## 2.5. Data collection

**2.5.1. Interview guide.** a semi-structured open-ended guide for Focus Group Discussions (FGDs) was utilized to facilitate comprehensive discussions. The guide, initially derived from prior studies [52, 53], was tailored to suit the specific context of the current study. It encompassed essential elements aimed at fostering a comfortable environment for participants to respond to sensitive inquiries and conclude the session with a sense of closure. The introductory section of the guide elaborated on the study's purpose, consent procedures, confidentiality assurances, and details regarding reimbursement. Questions within the guide were organized following the conceptual framework of Knowledge, Attitude, and Practice (KAP), utilizing the three PEN-3 model constructs as a theoretical foundation. The sequencing of questions followed a logical progression, including a neutral ice-breaker, transitional queries, and closing questions to signify the conclusion of the discussion while allowing participants another opportunity to share their viewpoints. The primary questions were structured in a similar manner, as outlined in Table 1. A designated time frame of 2 hours was allocated for the

**Table 1. Simple schematic structure of the FGDs guide, the sociocultural contexts of M among PLWHs, Gedeo-Zone, Sothern Ethiopia.**

| S.N | Topic | PEN-3 domain | Conceptual Framework |
|---|---|---|---|
| 1 | Personal source of knowledge | Cultural Identity | Knowledge |
| | Extended families role in knowledge acquisition | | |
| | Neighborhood role in knowledge acquisition | | |
| 2 | Personal perception of MS risks and long-term outcome | Cultural relationship and expectation | Attitude |
| | Enabler of personal attitude towards MS risks | | |
| | Nurturer of personal attitude towards MS risks | | |
| 3 | Positive (beneficial) practice held | Cultural empower | Practice |
| | Existential (neither beneficial nor harmful) practice held | | |
| | Negative (detrimental) practices held | | |

discussions following the pre-testing of the guide. These measures aimed to enhance the quality of the discussions and ensure that participants felt at ease and engaged throughout the session.

**2.5.2. Audio/visual recording and field notes.** The audio recording of all discussion was made by using a Huawei P7 smartphone digital sound recorder and was saved to a micro SD-card in the MP3 format and backed up daily to the lead author's laptop computer to prevent any unnecessary events. All the discussions were correspondingly recorded on video and picture using a Sony digital camera (Sony Lens, 5X Optical Zoom; 3,2–6,4/4,6–23). In which the files were stored on the SD-card, and then backed up to the researcher's computer as usual for similar reasons. Both used to recall the events. Besides, the field note on verbal and non-verbal cues were taken partly using a short way of nonfiction using a Sinar-Line notepad and Lexi-pen.

**2.5.3. Duration and data saturation.** Each discussion was lasts for one and half hours, and data saturation was discussed.

**2.5.4. Transcripts returned.** Finally, all the transcripts sent to the participants and their corresponding health workers took part in the study and then all the comments and corrections were secured.

## 2.6. Data analysis

**2.6.1. Number of data coders and description of the coding tree.** The data analysis procedure was iterative. In the analysis of the transcripts, themes were established using the PEN-3 model constructs to push the conclusions toward the conceptual framework of knowledge, attitude, and practices (KAP). The data coding process were undergone by two researchers, Dr. GirmaTenkolu (DGT) and Sadat Mohamed(SM). Both these coders were met several times and discussed to the extent to which categories reflected in the data and reached consensus on the segmentation, by providing a description of the coding tree, a summary text data formation, and partial determination of coding and code category. In the coding process, each data item has been given equal attention and codes have been thoroughly inclusive and comprehensive. All relevant extracts for each code have been collated and checked against each other and back to the original data set to maintain it to be internally coherent, consistent, and distinctive. Overall, enough time has been allocated to complete all phases of the analysis adequately without rushing a phase or giving it a once-over-lightly, until themes identified in advance or derived from the data.

Based on the individual and approaches used, it was completed into three phases. The first phase was iterative characterized with step by step activities of data capturing and transcription. It was accomplished with the integration of GT and SM manually using the pre-established codebook, which was constructed using the model as a theoretical basis and the KAP as conceptual frameworks A codebook is provided as other information (S2 Table). The 2nd phase of the analysis likewise lasted from data transformation to translation. Initially, all transcribed texts data existed in hard copy and Amharic language were soon transformed into soft data, with the help of two secretaries. Subsequently, translation of each document from its original language to English, accomplished sometime after all data collections were ended, with the assistance of a bachelor's degree holder English teacher on a foreign language from Cambridge University, in UK. In addition, the lead author was involved to cross-checking of the transformed and translated text data with its original document through taking of several samples.

**2.6.2. Software.** The third phase of the analysis was a step by step activity that ranges from uploading primary data into computer software's up to the synthesis and formulation of the themes. The primary data type created includes rich text document audio and video

recordings, picture and survey data on participants' demographic characteristics. The data collection procedure was iterative, comprising appraisals and discussions at various stages of data collection; continuous comparison of new data with previous data, and performing of the correct reforms on the questions. Moreover, reflections or hints related to the data in a memo were used by the author for future interviews. The documentation of all the originally formed physical data was accomplished in one place in a locked filing cabinet for each study site, using the large heavy-duty archival envelope per-event. Ultimately, all the primary soft data were uploaded into Atlas.ti7, version-7.5.7 software, as a primary documents family (PD), and then stored in a large electronic envelope called "Hermeneutic Unit." Additionally, SPSS version-22 software was used to handle and analyze the survey data.

**2.6.3. Derivation of themes.** The synthesis and theme formation process were made in five steps. Initially, reading of the text document several times, back and forth and understanding of the overall contexts of the transcripts were made. Subsequently, a summary of each transcripts formed were reviewed and color-coded by using the words, phrases, clauses, and sentences as a meaning unit to identify the knowledge, attitude, and practice themes. This was compared with initial coding done manually using a close codding method in the previous data analysis phases. Further review of the conceptual and theoretical frameworks revealed pattern not seen prior to analysis was done. Using the revealed connections, constraining the themes along the lines of the theoretical framework and the conceptual framework was made. Use of this method allows the lines of connection between the theoretical and conceptual frameworks more direct and clearer. This method was chosen because the research was grounded in theory and allowed for the extension of the theory into new areas of understanding. Coding the data along the major domains of the theory and then by sub-domain assisted in determining relationships of the sub domains through the major domains to the conceptual frameworks. Finally, the productions of the reports with the selection of vivid and compelling extract examples were made.

**2.6.4. Participants checking.** Finally, the report sent to the participants and health workers engaged in the study for verification give comment and to correcting it accordingly.

## 2.7. Trustworthiness

Data trustworthiness was upheld through a series of meticulous actions. This included pretesting the Focus Group Discussion (FGD) guide, establishing trust between the research team and participants before discussions, and thoroughly examining codes and categories post-discussion. Comparison of codes, categories, and main themes with interview data ensured the completeness of participants' narratives. Regular meetings among research members facilitated continuous revision of concepts and classifications derived from the data until a consensus was reached. Ensuring data trustworthiness also involved extensive time spent on reviewing the data, back-and-forth, and providing a detailed outline of the context and analytical steps. Additionally, categories and quotations from interviews were refined by various public health professional editors to enhance accuracy and clarity. These measures collectively contributed to the reliability and credibility of the data analysis process. Let me know if you need further modifications or additional information.

## 2.8. Research ethics and consent

The Helsinki Declaration and ethical standards and principles were considered in human research. The proposal underwent ethical clearance from the Addis Ababa University College of Health Sciences School of Public Health Research and Ethics Committee (REC) and the College of Health Sciences Institutional Review Board (IRB) with Meeting No.001/2017 and

protocol No.0069/16/SPH. Please see the ethical approval letter attached as other elements. An official letter from SPH was sent to health bureaus, institutions, and the Southern Nations Nationalities Regional health bureaus, Gedeo zone, and Woreda health bureaus, indicating the ethical approval reference number. Informed consent was obtained from study participants before commencing the study. The consent process included a detailed explanation of the study objectives, procedures, and potential risks and benefits. Participants provided written consent by signing a consent form to indicate their voluntary participation in the study. All consent forms were securely stored and can be made available for verification if required. Data collection was conducted anonymous, with no personal identification, and subjects were enrolled freely after receiving written clearance. Confidentiality was maintained throughout the study, with physical measurements conducted in a screened-off ART clinic room. Bio-chemical assays were conducted free of charge, and results were sent for further examination and treatment.

## 3. Results

### 3.1. Participants' characteristics

Thirty-two male and female respondents participated in the study. 68.8% (22) of the subjects were aged between 35 and 44, with an average age of 36.96 (±8.94) years. Among them, 40.6% (13) had been on antiretroviral therapy for 1–3 years, with an average duration of 5.2 (±3.16) years (Table 2).

### 3.2. Qualitative results

**3.2.1. Knowledge linked to health behavior.** In the cultural identity of the health behavior domain, how adults living with HIV (PLWHs), acquire health information about behavior was explored. The questions for this part focused on how knowledge was attained at the individual level. What role did extended family play in the course? How did structures and institutions found in the community played a role to gain knowledge of health behavior is the last part of this section. Indeed, while using this domain, special care was taken to recognize how did persons individually attained information on metabolic syndrome and its associated risks. All the quotations are verbatim without rectifications.

Accordingly, the study found that typical responses to the question "The long-term effects of ageing with HIV?" The participant's response under the following quotations stated:

*"So far, I haven't gotten ill. . .I even know people who have been using the prescription for a long time and have never had any health problems. As for me, I don't believe there will be any problems (W7_FGD1). "I don't think that taking the medication(ART drug) consistently will have bad side effects, but good benefits."(FP_Ind1)*

As participants response revealed that they often believe that regular use of prescription medication (ART) will not lead to negative side effects and provide good benefits. However, some participants disclosed as they have never heard of the general symptoms and causes of metabolic syndrome. The typical responses to the question "The long-term effects of ageing with HIV?" The term "metabolic syndrome" is often overlooked in discussions about the long-term effects of ageing with HIV, due to misconceptions about non-infected individuals being more susceptible to metabolic illnesses. The following statements echo this idea:

*"I had no idea of this disease existed(M2_FGD2)," "I was informed I had contracted the virus as a result of the test(M2_FGD2)," "I haven't felt sick so far (M8_FGD2," "I know folks who*

**Table 2. The socio-demographic characteristics and anthropometric measurements of the participants, in Gedeo zone, Sothern Ethiopia, 2018.**

| | Characteristics | No | % |
|---|---|---|---|
| **I** | **Socio-demographic** | | |
| **1** | **Type of discussion** | | |
| | In-depth interview | 2 | 33.3 |
| | Focus group | 4 | 66.7 |
| **2** | **Location** | | |
| | Dilla university referral hospital | 18 | 56.3 |
| | Wonago health centers | 14 | 43.8 |
| **3** | **Occupation** | | |
| | A counselor | 6 | 18.8 |
| | Carpenter | 1 | 3.1 |
| | Day labor | 7 | 21.9 |
| | Farmer | 6 | 18.8 |
| | Housewife | 9 | 28.1 |
| | Prisoner | 2 | 6.3 |
| | Private worker | 1 | 3.1 |
| **4** | Religion | | |
| | Muslim | 2 | 6.3 |
| | No religion | 2 | 6.3 |
| | Orthodox | 16 | 50.0 |
| | Protestant | 12 | 37.5 |
| **5** | Marital status | | |
| | Married | 22 | 68.8 |
| | Single | 1 | 3.1 |
| | Widowed | 9 | 28.1 |
| **6** | Educational status | No | % |
| | Illiterate | 8 | 25 |
| | Grade 1st to 6th | 7 | 21.9 |
| | Grade 7th to 612th | 16 | 50 |
| | Diploma and above | 1 | 3.13 |
| **7** | Area of residence | No | % |
| | Rural | 13 | 40.6 |
| | Urban | 19 | 59.4 |
| **8** | Age of the respondents in year | | |
| | $\leq$ 25 | 5 | 15.6 |
| | 26–34 | 5 | 15.6 |
| | 35–44 | 22 | 68.8 |
| **9** | Duration on ART in year | | |
| | 1–3 | 13 | 40.6 |
| | 4–6 | 10 | 31.3 |
| | 7–9 | 2 | 6.3 |
| | $\geq$ 10 years | 7 | 21.9 |
| **II** | **Anthropometric measurements** | **No** | **%** |
| **10** | **Systolic Blood Pressure** | | |
| | SBP $\leq$ 129 mmhg** | 24 | 75.0 |
| | SBP $\geq$ 130mmhg | 8 | 25.0 |
| | Total | 32 | 100.0 |

(*Continued*)

**Table 2.** (Continued)

| | Characteristics | No | % |
|---|---|---|---|
| 11 | **Diastolic Blood pressure** | | |
| | DBP </ = 84mmhg | 14 | 43.8 |
| | DBP >/ = 85mmhg | 18 | 56.3 |
| | Total | 32 | 100 |
| 12 | **Height** | | |
| | Height 1</ = 1.60 m | 4 | 12.5 |
| | Height 1.61 to 1.70 m | 27 | 84.4 |
| | Total | 31 | 96.9 |
| 13 | **Waist circumferences** | | |
| | WC </ = 88cm | | |
| | W C 88 to 101 cm | | |
| | WC >/ = 102 cm | | |
| | Total | | |
| 14 | **Weight** | | |
| | Weight 46–54 | 7 | 21.9 |
| | Weight 55–64 | 21 | 65.6 |
| | Total | 28 | 87.5 |
| 15 | **Body Mass Index** | | |
| | BMI</ = 20* | 8 | 25.0 |
| | BMI 21–24 | 2 | 6.3 |
| | BMI 25–30 | 8 | 25.0 |
| | Total | 18 | 56.3 |

FGDs: Focus Group Discussions

**mmHg: mill meter of mercury

*BMI: Body mass index

Ind: In-depth interview

FP: Female Patient

MP: Male Patient

*have been taking the drug (ART) for many years, yet, even people who bought the pill before it was delivered by the hospital. . .but I haven't seen them have any health problems as a result of using the medicine for so long(W7_FGD1)."*

Even when the participants were questioned about the term "metabolic syndrome," In response, some of the extracts from participants stated,

*"I was completely unaware of this illness. . . Once, inexplicably, I became ill. . . Consequently, I visited the nearby health center. . . Subsequently, I was informed that I had contracted the virus following the test. . ." (M8_FGD2) "Since starting the ART drug, I've had a severe headache: burning sensations like a fire. . . I feel like I have a wound in my head. . ." It turns me into an inebriated person. . .I know little about metabolic disorders. . ."(W3_FGD3) "My vision is blurry. . .I've had health issues that I didn't have before using the medicine (ART). According to my understanding, long-term usage of the drug (ART) may result in eyesight abnormalities (W2_FGD3)."*

The above extracts revealed that participants in a study were unaware of metabolic syndrome until they became ill and were informed they had contracted the virus. Since starting

the antiretroviral therapy (ART) drug, they have experienced severe headaches, inebriation, and blurred vision. They also mentioned having health issues they didn't have before using the drug, and believe that long-term usage may result in eyesight abnormalities. Some have witnessed people on ART for years without health consequences, but few reports of acquired an allergy, also known as "Shifitah" in Amharic that spreads over their entire face after starting the ART medication. Despite these symptoms, participants believe that continuing to live with the virus and using ART won't put them at risk for additional health problems.

Additionally, when asking insightful questions about every elements of the metabolic syndrome, quotes such as these

> *"In my view, high blood pressure is a health issue brought on by excessive blood flow. "Diabetic disease is a condition brought on by consuming lots of sweet things." An illness that developed in connection with diabetes is blood pressure. . . The presence of one implies the presence of the other. (W8_FGD1) "Sugar diseases have treatments, just like any other sickness" (M6_FGD2) highlight the issue. . ."(W3_FGD)*

As the aforementioned excerpts demonstrate, in casual conversation, the individuals' preferred term for metabolic syndrome was "hypertension and diabetes mellitus," with a reoccurring theme of "blood pressure and sugar disease." The text highlights the importance of blood pressure and sugar diseases, stating that high blood pressure is a health issue caused by excessive blood flow, while diabetes is a condition caused by excessive sugar consumption. Both diseases require special care and attention, and treatments are available for these conditions, as they are similar to any other illness. As can be seen below, a tiny percentage of people, however, answered this question in a different way.

> *"Actually, I'm having trouble remembering, but I believe I've heard something about metabolic issues. Even though she hasn't had any health issues as a result of using the medication (ART), I am aware that utilizing the medications (ART) for a prolonged period of time may cause fat redistribution or modification, the effect of which is an increase in physical size. Therefore, I believe that as PLWHs mature, there is a potential they will develop chronic diseases like high blood pressure and diabetes." (W7_FGD1) "I am aware that use of ART drug may also result in conditions like diabetes and high blood pressure."(FP_Ind1)*

Participants in a study expressed awareness of potential metabolic issues linked to living with HIV and taking antiretroviral therapy (ART). They mentioned concerns about fat redistribution, which could lead to physical changes and increased risk of chronic diseases like high blood pressure and diabetes as individuals age. They acknowledged the potential impact of HIV medications on metabolic health and suggested a proactive approach to managing their health and risks associated with living with HIV.

Moreover, the particular response to the question "What are the risk factors for MS and its components?" The following was said by participants:

> "Primarily, lack of physical exercise. . . as numerous ailments are more likely to affect an overweight body" *(W7_FGD1) "Diabetes and high blood pressures are problems caused by dietary abnormalities; in particular, eating meals often that contain meat might have negative health effects. I can comprehend it in this way (M8_FGD2)." (M6_FGD2). "Diabetes and high blood pressure have been associated with malnutrition (inadequate food intake). For instance, malnourishment could trigger numerous issues for a diabetic patient." (W8_FGD1)"Distortion in food intake may contribute to the progression of diabetes."(W1_FGD1)*

Participants noted that diabetes and high blood pressure are associated with unusual eating habits and dietary imbalances, such as meat-heavy diets and malnutrition. They stressed the importance of a healthy diet in preventing diseases like MS and emphasized the role of diet in managing diabetes. The seventh female participant in the first conversation reiterated the aforementioned circumstances, stating:

*"Primarily, lack of physical exercise. . . as numerous ailments is more likely to affect an overweight body" (W7_FGD1).*

The other second woman in the same discussion gave the following response to the above-mentioned probing questions:

*"When it is said that a person has HIV, high blood pressure, HIV, and obesity, people's perceptions may vary. However, those who have high blood pressure, diabetes, and other disorders. . .As a result of an excessively luxurious lifestyle, many problems arise." (W2_FGD1)*

The speaker emphasizes the importance of physical exercise for preventing various ailments, particularly those affecting an overweight body. They also highlight pointing out that leading a sedentary lifestyle or having excessive luxuries without maintaining a balance with physical activity can increase the risk of developing these disorders. By highlighting the negative effects of an overly luxurious lifestyle on health, the speaker is drawing attention to the importance of making healthy choices and engaging in physical exercise to prevent or manage conditions like high blood pressure and diabetes. This underscores the significance of lifestyle factors in maintaining overall health and well-being, especially for individuals who may already be at risk due to factors such as aging or living with HIV. Another lady in the third discussion responded in a unique manner to the given context.

*"Because we (PLWHs) frequently experience irritability and anger, we are more prone to developing metabolic issues like high blood pressure and diabetes." The negative beliefs about the virus and low quality of life may have influenced our behaviors (irritability and anger)." (W8_FGD3)*

A third participant noted that irritability and anger are linked to metabolic issues like high blood pressure and diabetes. These behaviors may be influenced by negative beliefs about the virus and a reduced quality of life, which affects their willingness to change behaviors to reduce health risks. The study emphasizes the need to understand the challenges faced by people living with HIV and implement interventions to address behavior changes for managing associated risks and complications. Whereas, the late comer lady participated in the first in-depth interview responded in the following manner.

*"I am aware that "Chinket" in Amharic and "tension" in English lead to high blood pressure, but I am uncertain about the cause of diabetes, for example. Nonetheless, I believe excessive consumption of sweets is to blame. I have mostly heard it put that way. (FP_Ind1)*

A woman at the first interview highlighted the connection between "Chinket" in Amharic and "tension" in English, indicating a potential link to metabolic syndrome, especially in adult PLWHs. The identification of "tension" as a new risk factor in the development of metabolic syndrome in this group was underscored. The study emphasizes the importance of considering how HIV infection can increase susceptibility to metabolic issues among PLWHs, integrating

HIV into personalized strategies for managing risks and preventing long-term consequences. Furthermore, the woman expressed uncertainty about the cause of diabetes, largely attributing it to excessive consumption of sweets. These results highlight the need to incorporate HIV status into intervention frameworks and provide health education to improve awareness of metabolic syndrome and lifestyle-related risk factors among PLWHs.

In response to the probing question, "What are the long-term effects of MS and its components?" while some participants given these responses. Furthermore, when asked about methods for controlling and preventing MS, participants expressed concerns regarding this topics.

*"I am convinced that individuals with high blood pressure or diabetes will eventually suffer from severe headaches, collapse, and eventually experience cardiac arrest leading to demise." (M6_FGD2)*

The other few reacted in this way,

*"If I have high blood pressure, the bulk of this is brought on by fat accumulation. . .Thus, blindness) and vision loss are implied."(W2_FGD1)"I am aware that utilizing the medications (ART) for a prolonged period of time may cause fat redistribution or modification. The effect of this circumstance is an increase in physical size. . .develops chronic diseases like hig blood pressure and diabetes. Our chance of developing heart disease is ultimately a result of these health issues. (W7_FGD1)*

Participants expressed varying views on the potential impacts of hypertension and diabetes. Some anticipated severe symptoms such as headaches, collapse, heart failure, and even death. Others associated fat accumulation with issues like blindness and vision loss. These show that there was a lack of understanding among participants regarding the long-term effects of MS and its components. They also highlighted the potential risks of chronic conditions like high blood pressure and diabetes, which could result in complications such as blindness and vision impairment. The responses indicated a solid grasp of the long-term effects of MS and its components, possibly influenced by personal experiences with prolonged ART use. The differing levels of comprehension among participants regarding these effects also revealed gaps in contextual knowledge among HIV-positive adults, emphasizing the importance of healthcare professionals providing comprehensive health education during follow-up care and conducting thorough mixed qualitative and quantitative studies in relevant regions. Further, participants showed concerns about controlling and preventing MS in a variety of ways when asked about this topic.

*"Diabetes patients should thus refrain from consuming such peas, wheat, raw meat, unfried/ unboiled eggs, and unpasteurized milk since they may contribute to HIV-related issues."(W1_FGD1) "I believe. . . blood pressure and sugar diseases that require extraordinary care and importance" (M6_FGD2)*

The 8th woman PLWH in the third discussion responded differently from the others, stating,

*"Because of this, I believe that avoiding such behavior (irritability and anger), via raising community awareness and improving overall quality of life, may be beneficial, for us to be safeguarded from such chronic disease, including MS." (W8_FGD3)*

Participants raised concerns about specific foods like peas, wheat, raw meat, eggs, and unpasteurized milk potentially worsening HIV-related issues. They recognized the influence

of diet on health outcomes, particularly in metabolic syndrome. Additionally, they emphasize the significance of controlling blood pressure and diabetes, underscoring the public health relevance of these aspects for overall well-being. Furthermore, other respondents proposed approaches to prevent and manage chronic diseases like MS, such as avoiding stress, enhancing community awareness, and enhancing quality of life. This viewpoint underscores the critical role of psychosocial factors and emotional well-being in overall health and disease prevention. They stress the interconnectedness of mental and physical health concerning MS, highlighting the importance of stress management and positive community involvement.

The study examined how HIV-positive adults seek health information, focusing on personal knowledge acquisition and the influence of extended family and community dynamics. Participants generally agreed that gaining awareness of metabolic syndrome (MS) risks and outcomes involved utilizing mass media (TV and radio), peer discussions, volunteer education, hearing about neighbor's illness tragedies, and other methods. Family members, especially spouses (both male and female), children, sisters, and brothers, were identified as the primary sources of education on MS risks. Some individuals received education due to specific misfortunes such as illness and the loss of family members. Various entities, including government structures, health-care systems, health experts, religious organizations, and the social group 'Idir,' employed diverse methods to spread information about reducing MS risk factors. Participant responses were analyzed to assess the contextual understanding of HIV-positive individuals, considering viewpoints from individuals, extended families, and neighbors. The results highlighted a lack of awareness regarding the potential impacts of HIV medications on metabolic syndrome and associated risks. There was a limited understanding of lifestyle factors that contribute to disease prevention, coupled with differing opinions on the long-term consequences of metabolic syndrome. The study stressed the importance of proactive health management by integrating HIV status into personalized intervention strategies. Additional quotes from participants can be seen in the attached supplementary information as (S1 Table). The following paragraphs describe in detail how the findings mentioned above were determined.

**3.2.2. Person.** The PEN-3 model suggests that an individual's cultural identity and connections to extended family and community traditions play a crucial role in the process of adapting to health education [21, 33–35]. The study on PLWHs overall found that personal experiences with serious illnesses, such as allergies, diarrhea, and swelling, can enhance health education. Mass media, such as radio and television, can serve as key sources of health information, enhancing overall health education. It emphasizes the importance of increased health education on MS risks and long-term effects, as well as behavioral adjustments in response to the illness. Mass media can be a valuable tool for educating individuals about MS risks and complications.

## 3.3. Personal experiences of severe illness tragedies as a form of health education

Interviewee response regarding how they educated about MS and its corresponding risks. A conjoint issue that emerged was "the personal serious disease tragedies were the colloquial words found as their source of health education." The under designated participant response informed the various efforts of the infectious and non-infectious disease events role to learn:

> "I experienced allergies, or 'Shifitah' in Amharic, as soon as I started taking the medication (W6_FGD3).""Initially, I had diarrhea. . .My face then began to swell. My situation became strange overall. . .I was completely involved in an unusual situation" (W7_FGD3).

Few quotations from participants indicated worry about the impact of chronic disease tragedies on education.

*"Seeing people die in an unexpected situation taught me about the dangers of these diseases" (W7_FGD1). I learnt more about these health issues after watching my neighbor become ill and require insulin (W7_FGD1).*

The interviewees discussed their education about MS and its risks, focusing on personal serious disease tragedies as a source of health education. They shared their experiences with allergies, diarrhea, and unusual situations, highlighting the importance of understanding infectious and non-infectious disease events. Some participants expressed concern about the impact of chronic disease tragedies on education, citing seeing people die in unexpected situations as a reminder of the dangers of these diseases. They also learned more about health issues after watching their neighbor become ill and require insulin.

## 3.4. The use of mass media as a source of health education

Participants were further questioned about "how they educated about MS and its corresponding risks?," Brief comments on the topic of " use of mass media " as important personal sources of health information for adults with HIV (PLWHs).

*"I get information through listening to a radio and watching television" (M6_FGD2).*

The study explores the role of mass media as a crucial source of health education for adults with HIV (PLWHs). Participants reported that they primarily obtain information through radio and television. This data aids in creating specific educational efforts to raise awareness, boost knowledge, and enhance health outcomes. It proposes using mass media in combination with other approaches to educate about MS risks and long-term issues, improving overall health education for MS individuals.

## 3.5. Peer discussion in the hospital and mass media as a form of health education

Furthermore, when addressing the query, "What is the primary source of health education?" participants provided responses.

*"Every morning, at the hospital, there is a group or peer discussion that provides us with health education. We discuss our health issues and relevant precautions, but metabolic problems are never addressed."(W1_FGD1)*

A woman in the first discussion noted that during morning hospital group sessions, health education is offered, but metabolic issues are not tackled. This suggests that the peer discussion lacks coverage of relevant topics, indicating a need for healthcare professionals to include education on MS and strategies for preventing and managing its long-term complications.

**3.5.1. Extended family.**   The PEN-3 model suggests that cultural identity and family connections are crucial for health education adaptation [21, 33–35]. The study emphasizes the importance of health education for people with PLWHs about MS, its risks, and long-term prevention methods. It highlights the role of spouses' spontaneous blood HIV testing, chronic disease incidents, and family advice. However, families often lack understanding of metabolic problems, risks, and prevention methods. Awareness from children, mothers, and friends can

help inform families about these issues. The study emphasizes the need for increased awareness about HIV, ART medication usage, and chronic conditions like diabetes and hypertension, emphasizing the importance of family education and support in addressing health issues and stress.

## 3.6. The extended family members played a role as a source of health education

A classic response to the question "for the first event that encourages the respondents who bothered for their health?" was "the spontaneous blood HIV tests that they (their family members) had undergone subsequent to the spouses' acquaintance with the virus" was the first event that encouraged the participant to bother about their health. This response confirms this concept.

> *"I learned about my health from my partner. . .When my wife became ill, she went to a medical facility for a test. . .and because she was discovered to have the virus, I was asked to do the test. . .Then we were both told that we, too, had the virus" (M4_FGD2).* "I found out because my spouse died of HIV. . .That's all there is to it. . ." (W4_FGD3) *"My spouse is my staunch supporter, advising me against consuming non-fresh food and drink, but we haven't talked metabolic health problems. . .To be honest, our awareness in this aspect is limited. (FP_Ind1).*

The speakers discuss their HIV status discovery through their partner's test results, highlighting their health concerns due to exposure to the virus. They mention their spouse's death from HIV and their limited awareness of metabolic health issues. Despite their spouse's encouragement, they have not raised these difficulties, emphasizing the need for further knowledge and awareness about metabolic health issues. The finding emphasizes the significance of recognizing one's own health and the impact of HIV on one's life. Participants also noted that the death of a loved one due to chronic conditions like diabetes and hypertension was a pivotal moment that raised concerns about their own health among individuals living with HIV. Here are a few quotes on chronic illness:

> *"I became acquainted with diabetes illness via a friend. . .This sickness affected her. . .She'd been informed she had diabetes and that she needed to take the medication properly. . .She stopped taking the medicine and died as a result of failing to act on the recommendation of her doctor" (W3_FGD1)."I have seen it on my mother. . . My mother is a diabetic patient. . . If she gets a minor wound, the wound never gets cured soon. . .She can't eat what she wants. . ."(W5_FGD3)*

The speaker discusses the importance of health education for chronic diseases like diabetes and blood pressure, emphasizing the role of family members, especially mothers and friends, in providing valuable insights into the risks and long-term effects of these conditions. They also highlight the role of exposure to hypertension and diabetes in providing health information to family members, especially through mothers and intimate acquaintances. Furthermore, this section will use participant responses to offer a quick summary of how the regular discussion made by the family members such as children, brothers and sisters are seen as sources of health education.

> *"My family is really supportive. . .They are frequently receptive to discussing concerns connected to my health problem (HIV and ART medication usage difficulties). . .My children, in particular, give me the medicine with the statement, "Drug taking time is up." They are really*

*beneficial in this regard. . .What if God hadn't sent them to me? (He waggles his index fingers at the sky)"(M8_FGD2). "Unquestionably, my family (my brothers and sisters) support me to have awareness" (W4_FGD1). "In terms of maintaining health and preventing sickness, we talk and teach each other in our families. . . In fact, we frequently concentrate on HIV-related topics (HIV and ART medication usage difficulties). . . . My family has no prior understanding of the topics we're talking. . .My children would not have hesitated to inform all members of the family if they had been aware of such things. . ."(M1_FGD4)*

The participants' family is supportive of their HIV and ART medication usage difficulties, especially their children's encouragement. They also support their awareness of HIV-related topics, focusing on health maintenance and prevention. However, their family lacks prior understanding of these topics, and their children may be more willing to inform their family about metabolic problems, risks, and prevention methods if they are aware of these issues.

**3.6.1. Neighborhood.** According to Airhihenbuwa, neighborhoods play a crucial role in shaping an individual's social environment, providing support or creating obstacles to health-related behaviors and decisions [21, 33–35]. This study explores social environments beyond the family that affect multiple sclerosis (MS) prevention and encourage risk behaviors among people living with HIV/AIDS (PLWHs). It highlights the crucial role of formal systems, especially government initiatives, in providing health knowledge through data collection and HIV/AIDS counseling. However, critics argue that media and radio information may not adequately address metabolic conditions. Informal sources, such as churches and social gatherings, are essential for individuals with HIV, and expanding these networks to include chronic diseases like MS could improve health awareness.

## 3.7. Formal organizational structures as a source of health information

The study revealed that formal systems, primarily government-linked, significantly influence personal health knowledge acquisition, with "government structure" being a crucial factor in collecting health information, underscoring the significance of formal organizational structures. In this sense, the data extract verifies the following:

*"Even if it isn't a direct recommendation," "The television program "Your Health at Home" 'Tenawo Be-Betwo' provides advice on HIV/AIDS, and it is hoped that government officials will implement measures to benefit the community "(W2_FGD1). "I learn more about health through other media, such as television and radio. . .We have lately heard health-related information from these media, despite the fact that it is not about such themes (Metabolic disease). . .And I read newspapers for health information on occasion. . ."(W7_FGD1).*

The television show "Your Health at Home" 'Tenawo Be-Betwo' provides HIV/AIDS counseling, presuming that government officials will adopt community-benefiting measures. They also reported that they learn about health from various media, such as television, radio, and newspapers, and considered them as important sources of health information. However, some participants argue that media, television, and radio health information is irrelevant to the metabolic condition.

Moreover, the study found the government system as a crucial supplier of health education, as evidenced by the following quotes.

*"In regard to government assistance for our health, the government supplies us with pills (ART medications), but no more assistance. . .On rare instances, health institutions distributed a nutrition item called as "Plamplate" in Amharic and a very healthy meal in English*

*that was regularly offered as a nutrition therapy to only those in need of sustenance, but it has since been discontinued. . ."(W5_FGD3)"Whereas, I got full support from health institutions. . .On top of that, health institutions are delivering information on various types of health problems in the outpatient department before issuing cards"(W1_FGD1)*

They stated that the government provides ART medicines for health, but no more aid is available. Health institutions occasionally deliver a nutritious meal called "Plamplate" in Amharic, which was once offered as a nutrition treatment. The "government system" refers to healthcare organizations' role in education and healthcare organizational structure, where medical facilities provide comprehensive health-related information before issuing cards. One responder perceived things in this light in respect to these settings.

*"They (government agencies) just talk about and teach about HIV. . .This is due to, I believe, the government's policy. . .It all comes down to HIV. . .Even on this topic (about MS, its risks and methods of prevention and controlling), it is not done. . .On the 21st of the century. . .every year. . .individuals recruited from sector offices are allocated to disseminate HIV information. . ."No one ever considered other related health issues" (W2_FGD3).*

He stated that the government agencies prioritize HIV due to policy, neglecting other health issues like MS. The rising prevalence of chronic diseases and metabolic syndromes calls for a rethinking of infectious disease policies to include PLWHSs and improve access to routine screening and management. Furthermore, the study reveal that various health professionals within healthcare organizations serve as sources of personal health knowledge acquisition, providing knowledge similar to organizational structures, as noted by individuals raising concerns.

*"In addition, health extension workers in each 'Kebele' have delivered HIV education. . . .In fact, we have never been informed about any other health issues (even metabolic health issues). . . ."(M1_FGD4) "The physicians (the doctors) advise us. . ."Heat everything you made before eating it, whether it's cabbage or something else. . . . "Never eat raw food. However, their advice had no link with the problem you said now (Metabolic syndrome)"(W2_FGD1)*

As stated in the preceding excerpt, Health extension workers are considered as health professionals who play a varying role in personal knowledge acquisition at the grass roots level. Doctors and other health practitioners also play a role in the dissemination of health information to them. Generally, this highlights the need of including the aforementioned health professionals in communicating health information about MS, its risks, and methods of prevention and control among PLWHs.

## 3.8. The informal structures as a source of health information

The informal systemic structures were also recognized by participants as an important primary source of health teaching. A reoccurring subject was "the churches." While a few people accepted the church's role without naming it, as seen by the responses:

*"The churches also teach us a lesson to protect ourselves and not transmit to others with negligence and not to commit sine. . .but they never teach us other health-related teachings. . .This form of instruction was only available to us through health care institutions." (W4_FGD1).*
*"The church also teaches us to be careful, not to blame others for our errors, and not to*

*sin. . .They never supply us with further health-related teachings, which we only acquire from health institutions. We're hosting a peer conversation to raise awareness. . ."(FP_Ind1)*

Other participants stated the following religious organizational functions as well as the name of the institution.

*"Coming to religious institutions; for example'Aboye-church' is doing its contribution by providing financial aid and awareness creation teachings about HIV. . ."(M8_FGD2)"Religious institutions (Churches), such as St. Mary and St. John churches, teach about HIV during Gospel preaching." They (the churches) emphasize how to stay healthy, stay with one spouse, and avoid adultery. "As for the diseases that would occur through time (like Metabolic syndrome); let alone for them to teach us, we have just heard of them from you" (M3_FGD2).*

Churches, while teaching caution and refraining from blaming others, do not provide health-related teachings. Instead, religious institutions like Aboye-church offer financial aid and HIV awareness. The study also identified the significance of informal systems as PLWHs' main resource for health information. It draws attention to the importance of social calls, or 'Idir' in Amharic, in helping people learn more about health issues. The quotes taken from the participants provided greater insight in this regard.

*"In this regard, I dare to say that members of 'Idir' have organized a discussion forum for health education during someone's death, by using the incident of deaths as a topic of discussion." If the occurrence is linked to HIV, the member would typically pay more attention to it and discuss it as a hot topic. And I believe that such a debate platform should be conducted on a long-term basis, with equal attention paid to the health issues of others, including the health problem linked with chronic disease."(W1_FGD1) "In addition, in social gatherings such as "Idir," we conduct peer discussion on health-related issues."(W6_FGD1).*

The respondent describes how 'Idir' members created a health education forum amid high mortality rates, particularly HIV-related cases. The results of the study suggest that since this social structure promotes group discussion and problem-solving, the platform should be expanded to address issues related to chronic diseases and provide a forum for people to discuss various health topics, particularly on the health issues related to MS, its risks, the method of prevention, and controlling of its long-term effects.

### 3.9. Attitude associated health behavior

The PEN-3 model evaluates health behaviors by examining the Relationship and Expectation (RE) domains, which encompass perceptions, enablers, and nurturers. This model begins with an analysis of how individuals' social circles affect their health knowledge, beliefs, and values (perception), followed by an exploration of enablers and nurturers. Enablers include resources and government roles, while nurturers are significant figures in a person's family, community, or organization who play a crucial role in shaping health beliefs, attitudes, and behaviors [21, 33–35].The study employed these concepts to investigate how the perceptions, beliefs, and values regarding metabolic syndrome (MS) risks and prevention, held by participants, their families, and their communities, influence their engagement with the topic. Health attitudes and perceptions are cultivated within these frameworks, highlighting the importance of influential individuals in one's social network. Findings reveal significant disparities in how People Living with HIV (PLWHs) view the risks associated with MS, with many deeming it less critical than HIV, perceived as a more severe condition. Sociocultural factors, including the perceived

severity of living with HIV, limited knowledge about MS, economic hardships, and insufficient support from non-governmental organizations (NGOs), were identified as factors that could either strengthen or undermine PLWHs' existing views on MS risks. Moreover, the study highlighted the importance of family support, the involvement of healthcare professionals, social stigma, and the actions of healthcare providers as essential sociocultural aspects that can either promote or obstruct the development of PLWHs' understanding of MS risks, risk management, and potential long-term effects.

Although not explicitly connected to the attitude component of the PEN-3 framework, individuals' perceptions of their surroundings and the impact of socially acceptable behaviors are fundamental to shaping attitudes toward the contexts studied. By amalgamating insights from the RE domains of the PEN-3 model with the attitude conceptual framework, this research illuminates how individuals living with HIV perceive and comprehend issues related to metabolic syndrome (MS). Despite some inconsistencies in the attitude dimension, the study highlights the significant role of social and cultural factors—such as personal characteristics, economic conditions, cultural norms, and organizational structures—in influencing the health behaviors of HIV-positive individuals. As a result, these sociocultural contexts contribute to the development of unfavorable perceptions and attitudes regarding the risks and prevention of metabolic syndrome within this demographic. The next paragraphs provide a detailed explanation of the findings mentioned above.

**3.9.1. Perception.**   The relationship and expectation (RE) domains of the PEN-3 model, as articulated by Airhihenbuwa, are designed to explore health-related attitudes by analyzing the effects of individual experiences, economic systems, cultural constructs, and institutional factors on these attitudes. The model begins with an examination of individual perceptions regarding the influence of their social circles on health knowledge, beliefs, and values, before moving on to the supportive and nurturing dimensions [21, 33–35]. This study specifically investigated how individuals' views on the risks, consequences, and preventive measures associated with metabolic syndrome (MS) affect their behaviors. Overall, the findings revealed notable discrepancies in how adults living with HIV (PLWHs) perceive the risks linked to metabolic syndrome, with many considering MS a minor health issue when compared to HIV, which is perceived as a far more serious condition. This suggests that the values and belief systems of PLWHs impact their understanding of both HIV and the potential risks of developing other health issues, particularly metabolic syndrome and its long-term repercussions. Furthermore, the study underscored the crucial role of sociocultural contexts—in particular, the perceived severity of living with HIV—along with factors such as insufficient contextual knowledge, familial influences, and organizational dynamics, in shaping the health behaviors of PLWHs. These factors contribute to negative perceptions and attitudes toward the risks associated with MS and its preventive measures.

## 3.10. Belief and values system

Participants systematically questioned who is responsible for health issues in general and the specific subject mentioned. They shared thoughts on how their beliefs and values surrounding MS risks, outcomes, and preventive measures influence their practices in chronic disease prevention.

*"One can only exist if one can care for oneself,""I am solely accountable for my well-being... I must prioritize self-care... What help can I expect if I neglect myself?" (M6_FGD4).*

They highlighted the importance of self-care and the repercussions of neglecting oneself. This highlights the participants' understanding of how personal beliefs and values impact their

actions related to chronic disease prevention, particularly in the context of Metabolic Syndrome (MS). It stresses the significance of empowering individuals to take an active role in their health through self-care practices and a sense of personal agency in disease prevention. Respondents also shared their thoughts on facing an MS issue, with one participant mentioning,

*"If I can manage a significant condition like HIV. . . other issues may not faze me." "If I'm informed about ailments such as hypertension and diabetes, they wouldn't bother me. . . because if I can cope with a major condition like HIV. . . other ailments wouldn't trouble me. . . I consider HIV to be more significant than these conditions. . ." (MP_Ind2).*

Respondents discussed managing significant conditions like HIV, indicating that if they can handle it, other issues may not be as daunting. Some compared HIV to other metabolic ailments, viewing it as more critical. While some noted that HIV only impacts symptoms, their commitment to antiretroviral therapy (ART) counters this view, showing resilience and acceptance. These varied responses suggest individuals facing MS and HIV have complex outlooks on managing chronic health conditions, revealing the subjective nature of perceiving illness and the challenges based on health contexts. Successfully managing HIV can alleviate concerns about other conditions, showing the ripple effects of health management. HIV can shape one's values and healthcare perspective, reflecting on how chronic illness influences experiences and priorities. Persistence through ART demonstrates resilience in navigating life with a chronic condition, highlighting the intersection of physical health, emotional well-being, and values in their journeys. Other participants also shared insights on the mentioned conditions.

*"My only recourse is to live through medication (ART). . . (grinning as if swearing. . . then. . . smiling. . . now). . ." (FP_Ind1) "Chronic diseases pose severe health challenges. . . It's daunting to even contemplate these ailments, let alone metabolic issues. . . Once afflicted with these conditions, the specter of death looms ever closer. . . It might eventually force me to leave my children as orphans. . . What could be more heart-wrenching than leaving them behind due to death?" (W7_FGD3).*

Additionally, participants discussed risk-reduction techniques and managing chronic illnesses like metabolic syndrome (MS).

*"We have abandoned thinking about health." "I, too, see it like others do. . . Poverty, unemployment, and other social ills are prevalent. . . As a result, many have stopped worrying about their health. . . Everyone here is racing to win their daily bread. . . As for me, I don't see it as an issue. . . The real issue is a lack of food" (M5_FGD4). "No disease is big (major) and no disease is small (minor). . . Generally, every disease is harmful. Likewise, I think these health problems (metabolic diseases) need due attention. . . It's because. . . If these once happened to come, we will be forced to purchase more drugs and use. . . we will be obliged not to eat food item we have at hand. . . Instead, we will be obliged to select food item risks. . . And, this is unaffordable for us" (M3_FGD4).*

Many participants expressed how social issues like poverty and unemployment cause them to overlook their health. They emphasized a lack of food as the main concern and believed that no disease should be taken lightly. They stressed the importance of addressing metabolic diseases as neglecting them could lead to increased medication costs and dietary restrictions, which are unaffordable for many. This highlights the urgent need for comprehensive solutions

addressing not just the medical aspects of chronic diseases like metabolic syndrome in adults living with HIV but also the underlying socioeconomic factors affecting health outcomes. It underscores the necessity of holistic interventions prioritizing food security, affordable health-care access, and support for individuals facing challenges such as poverty and unemployment. By acknowledging broader social determinants of health, efforts can be made to create a more equitable and supportive environment for individuals managing complex health conditions. Concerns about chronic diseases like MS were deeply felt among participants, impacting their lives and their loved ones, particularly their children. The fear of abandonment due to illness and the prospect of death are significant sources of distress. This sheds light on the emotional burden and difficult decisions individuals with chronic illnesses must face, like managing their health with medication and navigating an uncertain future. These shared sentiments under-score the importance of support systems for both medical and emotional well-being in condi-tions like MS. It further emphasizes the need to address the psychosocial aspects of illness, providing coping strategies for managing associated risks and challenges. Ultimately, it high-lights the profound impact chronic illness can have on individuals and their families, under-scoring the importance of comprehensive care and empathy in handling such conditions.

Furthermore, in the realm of beliefs and values held by adults living with HIV (PLWHs) regarding Metabolic Syndrome (MS), participants were also questioned regarding preventive strategies related to diet, physical activity, and managing harmful substance use habits. Com-monly noted limitations included "limited strengths," challenges in accessing protein-rich foods necessary for a balanced diet as conventionally perceived, and the constraints of being employed as a day laborer, as outlined in participant statements.

*"It's beneficial to play sports. . .It's not going to work out for me, though.I am a day laborer, therefore to begin with, I don't have the strength to finish. . .Everyone knows that we need to consume enough nourishment, such raw meat, in order to play sports.To make this happen, we'll need money.Because of my financial circumstances and health, I am unable to do so"* (M1 FGD4).

Somebody else said,

"Most of the time we. . .The fact doesn't need to be hidden. Due to the illness, we are regu-larly bothered. We might so periodically light up a cigarette or have a drink in order to for-get the frustration. It also raises another problem, as we are aware, but we repeat the process" (M4_FGD4).

The individuals conveyed their wish to participate in sports because of the advantages that come with it, but they also emphasized the challenge that comes with working as a day laborer and not having the necessary stamina, money, or nourishment. They also stated using alcohol or cigarettes as a coping method to get through the difficulties brought on by their medical condition. The responses from participants highlight the significant hurdles faced by PLWHs in adopting healthy lifestyle practices, especially in terms of dietary choices, physical activity, and managing substance use. These challenges often stem from physical limitations resulting from opportunistic infections and daily medication, compounded by economic hardships such as food insecurity and low wages. The study underscores the struggles of PLWHs in maintaining a nutritious diet due to limited access to essential foods, financial constraints, and work demands. Addressing these socioeconomic disparities and providing support services is crucial to ensure that individuals can access the necessary nutrients. Additionally, while sports can be beneficial in preventing metabolic issues, PLWHs encountering health and financial

barriers may find it challenging to participate in physical activities. The study underscores the importance of tailored, accessible, and affordable physical programs that cater to their specific needs. Furthermore, the study sheds light on how some PLWHs resort to smoking or drinking as a way to cope with their health challenges, underscoring the importance of considering mental health and emotional well-being in holistic care approaches. Recognizing the perspectives and values of PLWHs is crucial for designing effective strategies to prevent metabolic problems. Customized interventions can help promote healthier lifestyle choices and reduce the risks associated with metabolic complications.

**3.10.1. Enablers.** According to the PEN-3 model's Relationship and Expectation (RE) domains, enablers include resource accessibility, availability, government officials, managers, referrals, skills, and service types, all of which can either facilitate or hinder change [21, 33–35]. This study emphasizes the importance of support systems from healthcare organizations and government involvement in promoting health-conscious behaviors among adults living with HIV (PLWHs) regarding Metabolic Syndrome (MS). It identifies positive influences on diet, physical activity, and substance use, as well as the role of non-governmental organizations in shaping MS risks and preventive measures. However, it also highlights significant challenges, such as a lack of contextual knowledge and standardized actions, which adversely affect MS prevention for PLWHs. Participants voiced concerns about insufficient attention from healthcare and governmental entities and socioeconomic factors that deter proactive engagement in MS risk reduction. The study advocates for a holistic approach to effectively support adults managing lifestyle-related risks, stressing the need for coordinated efforts to improve perceptions and attitudes towards MS prevention in this population.

## 3.11. Enhancing enablers

Participants expressed concerns about the role of healthcare organizations in improving support systems, particularly in providing education and assistance to PLWHSs, and their current stance on government support, even though the term "government" was not explicitly mentioned.

> *"I received full support from healthcare institutions" (W1_FGD1). "Regarding our health issue (HIV/AIDS). . .the government supplies us with the tablet, but it doesn't offer any other support" (W5_FGD3). "*

The speaker expresses dissatisfaction with the government's efforts to raise awareness about HIV/AIDS, noting that while medication is provided, additional support for metabolic issues and associated risks is lacking. The significance of organizational roles is underscored in influencing attitudes towards lifestyle-related risk reduction strategies for MS. It indicates a need for healthcare organizations to reassess their strategies and services to better meet the needs of PLWHs, particularly in terms of addressing lifestyle-related risks and preventive measures. The term "government" is used ambiguously to refer to the role of healthcare organizations in enhancing education and support for People Living with HIV (PLHIV), without explicitly naming any specific entity.

Building on this, we inquired, "Have there been any structural or systemic efforts made to facilitate or hinder the prevention of the syndrome?" Other respondents joined the discussion as it progressed.

> *"As we often see, the government diligently works to raise awareness on important issues within the community through various means. . ." (M8_FGD2).*

Subsequently, we posed the question, "Have any structural or systemic efforts been implemented to enhance or impede the prevention of the syndrome?" A number of additional participants shared their views.

*"As is commonly known, the government consistently invests significant efforts in promoting community awareness on critical issues" (M3_FGD2).*

Participants noted that the government is actively increasing public awareness about crucial issues and addressing them through various programs, potentially strengthening their ongoing endeavors to combat metabolic diseases. Participants generally highlighted that the support systems of healthcare organizations and government involvement were seen as factors that enhanced enabling conditions for adults living with HIV in their attitudes towards Metabolic Syndrome (MS) and in adopting health-conscious behaviors, particularly regarding diet, physical activity, and substance use. The finding underscores the need for a coordinated approach to address lifestyle-related risks of metabolic syndrome among adults living with HIV. Enhancing education, assistance, and government support can improve well-being, quality of life, and positively influence attitudes towards preventive strategies and healthier lifestyle choices.

The study also revealed that specific non-governmental organizations significantly enhance individuals' understanding of the relevant circumstances, as described by a female patient.

*"The OSA organization has been instrumental in raising my awareness," noted a female participant, "this organization focuses specifically on HIV-related issues, greatly enhancing my understanding" (FP_Ind1).*

She indicates that non-governmental organizations like OSA significantly heighten awareness of HIV-related matters, potentially increasing involvement in metabolic disorders. In essence, the study found non-governmental organizations role played as enhancing enablers in influencing perceptions of MS risks and developing effective preventive and control measures.

## 3.12. Discouraging enablers

On the contrary, participants expressed concern about enablers' negative impact on preventing the syndrome due to a lack of contextual knowledge and system-level standardized actions. Some respondents shared the following:

*"The lack of adequate understanding and the absence of the needed activities. . .due to inadequate attention provided by all responsible entities, affects people, families, and government bodies. . .I suppose this is a disadvantage" (M7_FGD2). "I believe it is due to the lack of consistency in solid activities carried out by the government and non-governmental organizations to alleviate these problems (MS) and other health issues" (M3_FGD4).*

Participants reported inadequate attention by responsible entities, affecting people, families, and government bodies due to inconsistent activities in addressing health problems and MS. The inadequate attention from responsible entities, including healthcare organizations, government bodies, and non-governmental organizations, can have a direct impact on individuals living with HIV and their families. Without proper support, education, and resources to address lifestyle-related risks associated with metabolic syndrome, individuals may face challenges in managing their health effectively. This lack of attention can lead to negative health

outcomes and reduced quality of life for PLWHs and their families. They further disclosed that the current abandoned function of NGOs is hindering the enablers of the study theme. This notion is represented in the participant responses below, as follows:

*"Currently, there is no organization engaged in providing health-related aids and awareness creation education in the area. There were some the prior time. For example, there was a group named the "Medan Act". . .It was extinguished a long time ago" (W2_FGD3).*

The other provided as

*"In this aspect, my perspective is a bit different. . .because when we are living with the virus (HIV), we focus on activities to be done in order to end the virus. . .So, we often do not consider such issues. . ." (W3_FGD1) "HIV is so dreadful to me" (M2_FGD2), "Following this (HIV), you may be exposed to many and varied problems; particularly, lack of money would inevitably challenge you" (M6_FGD2), "You will especially be troubled not to work your job properly" (M6_FGD4), as well as "I regard my living with the virus as a big hindrance" (M2_FGD2).*

The responses highlight the significant challenges faced by individuals living with HIV/ AIDS, including financial, work, and job barriers, often neglecting other issues such as job performance and dependence. The absence of consistent activities by government and non-governmental organizations to address health problems and metabolic syndrome signifies a gap in the overall healthcare system's responsiveness to the specific needs of PLWHs. Governments and organizations have a responsibility to implement policies, programs, and interventions that support PLWHs in managing their health holistically, including addressing metabolic syndrome risks. The lack of consistent efforts in this area can hinder progress in improving health outcomes and overall well-being among PLWHs. As seen in the quotations below, participants in the discussion likewise attributed their socioeconomic background to a significant negative impact on their current perception of MS and its associated risks.

*"I perceive our being dependent (needy); our being illiterate; our being daily laborers for winning our daily bread; as well as having a meager income, as a hindrance" (M3_FGD4). "As is well known, the population number is greatly expanded here. . .there is insufficient farming land in comparison to the population size in our Zone. . .As a result, we are unable to provide alternative food production to support ourselves and our family" (M4_FGD4). "The absence of food aid here is the foremost problem that hinders us not to improve our feeding style" (M2_FGD4).*

They reveal that a region's population is dependent, illiterate, and daily laborers, with meager incomes hindering their needs. Despite schools, many youth choose daily labor jobs, and limited farming land availability.

**3.12.1. Nurturers.** Per the Relationship and Expectation (RE) domains of the PEN-3 model, a Nurturer encompasses individuals in the extended family, friends, and peers who influence health beliefs, attitudes, and actions [21, 33–35]. The study indicates the significant role of family support, community experiences, and health professionals in fostering positive perceptions among adults living with HIV (PLWHs) regarding metabolic syndrome (MS) and its risk reduction methods. However, challenges such as inadequate knowledge, social stigma, and healthcare system limitations hinder family members from effectively nurturing these perceptions.

### 3.13. Enhanced nurture

As per the recommendations of caregivers, people living with HIV (PLWHS) engage in specific health behaviors. One respondent expressed the following sentiment:

*"I value my family's support, particularly from my spouse and children. They help me in various ways, such as reminding me to take medication on time. Additionally, they provide valuable insights on virus-related concerns. My family's support, including that of my spouse and children, is truly strong." (M1_FGD4)*

The speaker conveys appreciation for the support of their spouse and children. This highlights the significant influence of family members, such as spouses, children, and mothers, in promoting healthy living and shaping health beliefs among PLWHs. The support provided by family members in terms of medication reminders and insights on virus-related concerns not only helps individuals with HIV manage their health effectively but also contributes to their overall well-being. This underscores the importance of social support networks in encouraging positive health behaviors and attitudes. In a related context, other participants also shared their thoughts:

*"I gained knowledge about chronic illnesses from my mother, who is diabetic. Witnessing her struggles has made me realize the gravity of such health issues." (W5_FGD3)*

Participants recounted their encounters with illnesses, underscoring the pivotal role of family members in promoting healthy living and shaping health beliefs, particularly the influence of children, mothers, and spouses. The experiences of family members, such as a mother who is diabetic, can contribute to PLWHs' understanding of chronic illnesses and health management strategies. By sharing their own health journeys and insights, family members can play a vital role in shaping the attitudes and behaviors of individuals living with HIV towards preventive health measures, including managing metabolic syndrome risks. This interconnectedness of health experiences within families highlights the potential for mutual support and learning opportunities. Respondents' subsequent responses may best illustrate the community's deepened familiarity with illness tragedies, which has shaped their current perceptions of MS risks.

*"I became more aware of these health issues (chronic disease) after witnessing my neighbor's illness and use of insulin" (W2_FGD1). "Seeing people succumb to sudden situations made me realize the severity of these diseases (metabolic issues)" (W7_FGD1).*

The participants' health outlook is influenced by their comprehension of diseases, their neighbors' ailments, insulin usage, and sudden fatalities, with healthcare professionals often prioritizing the deterioration of HIV infection over long-term MS symptom management. The findings underscore the importance of considering family dynamics and relationships in promoting wellness and lifestyle-related risk prevention among PLWHs. Family members, including spouses, children, and mothers, not only provide practical support but also emotional and informational support that can positively impact individuals' health beliefs and behaviors. Recognizing the crucial role of family support in shaping attitudes towards health can inform targeted interventions and support strategies that consider the interconnectedness of individual and familial health outcomes.

Participants also disclosed that healthcare personnel have an augmented nurturing role.

*"The doctors advise us. . . Regarding diet, they caution against consuming uncooked food"* (M4_FGD2). *"Health workers offer me strong support. . . They frequently notify us in advance about administering medications. . . Additionally, they advise us to break unhealthy habits, such as smoking and drinking"* (M3_FGD4).

According to the excerpts, healthcare personnel play a vital role in nurturing patients by advising against uncooked food, supporting medication administration, promoting healthy habits, and providing appropriate dietary guidance. They educated individuals on HIV/AIDS, emphasizing the necessity of heating food before consumption, avoiding raw foods, and cautioning against consuming prepared foods. The study highlights the growing importance of various healthcare providers in shaping perceptions of MS risks, prevention, and long-term implications. By engaging more with healthcare professionals, People Living with HIV (PLWH) can enhance their understanding of mitigating MS risks and managing long-term consequences.

### 3.14. Discouraging nurture

The family's role in shaping attitudes towards MS risk reduction and managing its long-term impacts appears to be discouraging, as indicated by the following quotes.

*"My family lives far away. . . They don't support me"* (W2_FGD1). *"I want to emphasize that my family didn't support me. . . I can't blame anyone but myself, you know why? Because I chose not to inform them!. . . I did it intentionally. . ."* (W6_FGD1).

As the discussion reveals, while the vital role of family support, community experiences, and health professionals in nurturing positive perceptions of PLWHs towards MS and its risk reduction methods, but the challenges faced by family members due to lack of knowledge, social stigma, and inadequate healthcare systems discourage in nurturing their perceptions. The insights from the discussion underscore the significant impact of family support, community experiences, and healthcare professionals in shaping the perceptions of people living with HIV (PLWH) towards metabolic syndrome (MS) and its associated risk reduction methods. These sources of support play a crucial role in providing education, guidance, and encouragement for individuals to adopt positive attitudes and behaviors towards managing their health effectively. A male participant further emphasized this, stating,

*"My family doesn't want to acknowledge my problem, let alone help me. . . I work to make a living. No one lends me a hand"* (M1_FGD4).

He said that his family doesn't support him or acknowledge his health problem. He disclosed that he works for a living and that nobody helps him. Another woman expressed,

*"I have no support. My in-laws are particularly responsible for my current health condition, besides not being able to support me (with a deep sorrow reflecting on her pale face)." "They (the family) were unaware of metabolic challenges. If they were, they could have guided me"* (W4_FGD1).

She felt that her in-laws are responsible for her current health condition and was unaware of metabolic challenges. The experiences shared by these participants highlight the significant impact of lack of family support and understanding on the perceptions and attitudes of adults living with HIV (PLWH) towards lifestyle-related risks of metabolic syndrome (MS) and

prevention strategies. These experiences underscore the need for a more empathetic and supportive environment that addresses the emotional, practical, and educational needs of individuals living with HIV. By fostering understanding, empathy, and collaboration within families and communities, it is possible to promote positive attitudes, empower individuals to make informed decisions about their health, and improve overall well-being among PLWH facing metabolic syndrome and related health risks.

One of the overarching issues emerged was about the negative impact of a family unit's lack of understanding and negative perspective on self-exposure on their health. In the second discussion, a woman exhibited a unique response to this idea.

*"Every community varies...Some are educated, and some are uneducated...As compared to these,....The uneducated community rarely does so in such activities...To your surprise, we come across people who do not have any know-how about HIV...Imagine! How many years have gone out since HIV was introduced into our country? What does this indicate? It indicates that the activities being carried out are insufficient."(W2_FGD3).*

The response reveals knowledge gap between educated and uneducated communities, especially concerning HIV, not only points to insufficient prevention efforts but also indicates a negative perception of the risks associated with metabolic syndrome. This disparity in understanding and awareness can lead to suboptimal health behaviors and outcomes among individuals living with HIV. It underscores the urgent need for improved education and awareness campaigns that target not only the general population but also specifically addresses the unique challenges faced by individuals living with HIV. By bridging the knowledge gap and promoting a better understanding of the risks associated with metabolic syndrome, these efforts can help empower individuals to make informed decisions about their health and adopt preventive strategies effectively.

Additionally, the research revealed that community stigma and discrimination significantly impede the adoption of healthy habits and their impact on individuals living with PLWHSs.

*"It's pointless to talk about the community (gesturing)... What I've found is that the community strongly disapproves of the utensils we use... Despite all our efforts, the community continued to stigmatize or discriminate against us in social situations. (With a pained expression on her face)... What surprises me (pointing her finger at herself)... They discriminate not only against us but also against our children... They don't allow their kids to play with ours... My child is undergoing ART just like me... When my child interacts with other children, the community often insists that they shouldn't play with my child... Knowing this, I tell my child not to play with them... And my child, in turn, wonders, "Why do they say this?" and "Why can't I play with them?" That's why I don't let him... (Appearing distressed and despondent... as tears well up in her eyes, her words trail off... Wiping away her tears, everyone in the room fell silent for a few minutes)... (After a while, she finished her sentence); the community is causing me distress..."(W4_FGD1).*

People living with HIV faced considerable pressure from community-driven discrimination, exacerbating their health conditions and impacting their children. This stigmatization resulted in a sense of isolation and an inability to seek support, making it challenging for them to deal with these difficult circumstances. The implications of this stigma and discrimination are far-reaching. Firstly, it severely hampers efforts to promote healthy behaviors and preventive strategies among individuals living with HIV. The negative attitudes and actions of the community not only undermine individuals' confidence in managing their health but also

contribute to a sense of shame and isolation, making it difficult for them to access the necessary support systems. Moreover, the distress caused by community stigma and discrimination can have detrimental effects on the mental and emotional well-being of those living with HIV. The sense of being ostracized and judged can lead to increased stress, anxiety, and depression, further complicating the management of both HIV and metabolic syndrome. Addressing this issue requires a multi-faceted approach that involves community education, advocacy for inclusivity and acceptance, and the promotion of empathy and understanding. By challenging stigma and discrimination, communities can create a more supportive environment that empowers individuals living with HIV to prioritize their health, seek necessary support, and engage in positive lifestyle behaviors to mitigate the risks associated with metabolic syndrome. Ultimately, fostering a culture of acceptance and compassion is essential in promoting the well-being and health outcomes of individuals living with HIV.

Furthermore the study highlighted the inadequate role of health professionals in discouraging the development of perceptions and attitudes of adults living with HIV (PLWHS) towards lifestyle-related risks of metabolic syndrome (MS) and prevention strategies, as revealed by interview responses.

*"To be frank, the healthcare providers never explicitly inform us whether our ailments are related to the medication or not. . . They (the healthcare workers) just provide us with the standard treatment. . . It would have been better if they had helped us in raising awareness, especially about metabolic illnesses, as you've just informed us. . . When we came in for medication, they never offered us any guidance on our dietary concerns. . ."(W2_FGD3).*

Interview findings indicate that healthcare providers often do not communicate the connection between patients' conditions and medication, nor do they address dietary issues during medication visits. Specifically, the research shows that care for metabolic syndrome (MS) in HIV patients is impeded by healthcare workers' inadequate provision of education on the condition, particularly concerning dietary matters. Insufficient support leads to a limited awareness of lifestyle risks linked to MS. The study stresses the importance of healthcare professionals enhancing patient education, offering lifestyle advice, and delivering holistic care to effectively manage health and lower MS risks. Improved patient education, personalized guidance, and patient-focused care can result in enhanced health outcomes and informed decision-making. In the fourth discussion, a man echoed similar sentiments.

*"Though we have health extension workers in our community. . . Their focus is solely on maternal and child health, as well as HIV and TB patients. . . Undoubtedly, it's commendable in its own right. . . But they didn't do the same for other health issues. . . So I wouldn't say they helped us with other health concerns."(M5_FGD4).*

Participants highlight the negative impact of government structures, such as the lack of medical professional involvement, on the knowledge and attitudes of people living with HIV towards metabolic syndrome risk reduction and management. When healthcare providers neglect the specific needs of HIV patients with metabolic syndrome, gaps in care and missed preventive opportunities can arise. This underscores the critical need for a holistic approach to healthcare for those with HIV, emphasizing the importance of healthcare workers receiving proper training and resources to educate patients on metabolic syndrome, dietary adjustments, and lifestyle changes. Governments should prioritize involving medical professionals in addressing the unique health needs of HIV patients, including metabolic syndrome prevention and management. Additionally, they note the limited focus of health extension workers in

Ethiopia on maternal and child health issues, potentially worsening the challenges for those with HIV in accessing comprehensive care for metabolic syndrome and other health concerns. Expanding the role of health extension workers to cover a wider range of health issues is crucial to ensure that individuals with HIV receive holistic care that meets their diverse healthcare needs. By bridging these gaps in healthcare delivery and support systems, the overall health outcomes and quality of life for individuals with HIV and metabolic syndrome can be enhanced.

### 3.15. Practice associated behavior

Airhihenbuwa proposed that the cultural empowerment (CE) domain of the PEN-3 model is intrinsically linked to health behavior practices. CE emphasizes the influence of sociocultural contexts on individual lifestyle choices, which can be classified into positive, existential, and negative habits. Positive practices enhance health, while existential practices may have mixed effects. Conversely, negative practices often result in adverse health outcomes and reflect harmful behaviors, prompting a shift in focus from self-blame to alternative frameworks for understanding health [21, 33–35]. This study explores the sub-domains of positive, existential, and negative practices, examining the conceptual foundations of these behaviors. Overall, our findings revealed participants' responses within the empowerment health behavior domain, encompassing positive, existential, and negative practices. Notably, positive behaviors were clearly articulated. Participants identified habits such as consuming fruits and vegetables, limiting harmful substance use, and engaging in intentional physical activity as positive practices that people living with HIV (PLWHs) perceived as measures to mitigate the risks of metabolic syndrome (MS). Moreover, the longstanding dietary practices of consuming 'Kochoo', 'Bulla', 'Injera', accompanied by 'Wot', emerged as existential practices that are culturally regarded as beneficial, despite the lack of scientific evidence supporting their impact on the progression of MS. In contrast, some participants acknowledged negative practices, including the consumption of raw meat, a sedentary lifestyle, and the use of locally produced alcoholic beverages, such as 'Areke,' 'Tella,' and 'Tejj,' along with cigarette smoking. The use of a substance known in Amharic as 'Khat' was also identified as a negative practice. The findings suggest that while many of the negative practices prevalent in the community are culturally accepted habits, there is a pressing need to address and mitigate their health risks.

Additionally, as outlined in Airhihenbuwa's model, the cultural empowerment (CE) domain of the PEN-3 model is significantly associated with health behavior practices [21, 33–35]. The findings from this study reveal that while adults living with HIV (PLWHs) recognize the health advantages of consuming fruits and vegetables, they lack specific guidance on the appropriate frequency and portion sizes necessary for effective health maintenance and the prevention of metabolic syndrome. This gap underscores a disconnect between their awareness of healthy eating habits and their actual dietary practices. Furthermore, the study highlights notable deficiencies in health preventive measures and knowledge related to diet, physical activity, and substance use, indicating an urgent need for more effective strategies to address and manage metabolic syndrome (MS). The Fig 1 attached as separately, together with the supplemental data file attached as (S3 Table), provide participant quotes, research inquiries, a conceptual framework, and study question responses.

**3.15.1. Positive practices.** Airhihenbuwa characterizes positive practices as those deeply rooted in individuals' beliefs and knowledge, recognized for their role in promoting health within the cultural empowerment (CE) component of the PEN-3 paradigm [21, 33–35]. The study broadly indicates that the consumption of fruits and vegetables is beneficial for adults living with HIV (PLWHs). Incorporating vegetables such as tomatoes, sweet potatoes,

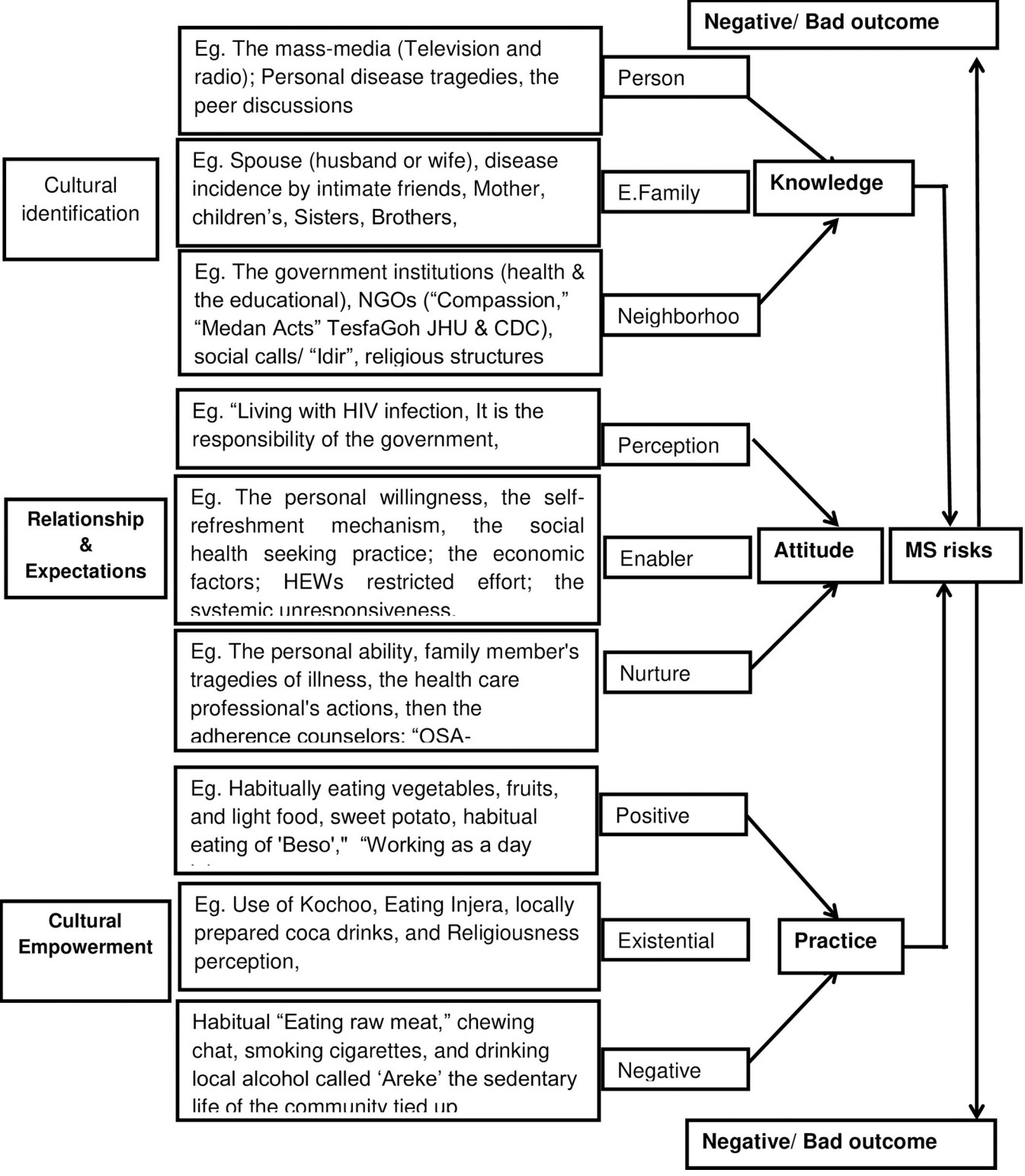

**Fig 1. A schematic summary of PLWHSs comments.**

potatoes, carrots, Swiss chard, cabbage, and pimento, alongside fruits like avocado, banana, and mango, can be crucial in preventing metabolic syndrome (MS) and the associated health risks for this demographic. Moreover, some PLWHs have identified reduced salt intake as a valuable practice for health maintenance. The study also highlights the significance of regular physical activity, discouraging sedentary behavior, and promoting engagement in indoor activities as essential habits that can diminish the risk of developing metabolic syndrome among adults with HIV. This underscores the importance of adopting a holistic lifestyle that integrates physical activity and indoor pursuits to enhance the health and well-being of this population. The findings outlined above will be elaborated upon in the subsequent paragraphs.

## 3.16. Maintaining healthy eating habits

According to the study review, while some respondents shared the following positive feeding habits:

*"I believe that if possible, incorporating the following foods into our regular diet—vegetables, tomatoes, sweet potatoes, potatoes, carrots, tomatoes, Swiss chard, cabbage, and pimento— might help lower our risk of developing such health issues as MS and its long-term effects. (W5_FGD1)*

Another few participants' response reveals

*"Consuming fruits, avocado, mango, and light, non-discomforting food, while enhancing our overall dietary habits, can help us steer clear of related health issues." (W8_FGD1)*

The respondents believe that incorporating specific foods like vegetables (including tomatoes, sweet potatoes, carrots, Swiss chard, cabbage, pimento), fruits (like avocado, mango), and light, non-discomforting foods can have preventive effects on health issues. These foods are rich in nutrients that are beneficial for overall health and may help in lowering the risk of developing certain conditions. The study highlights the importance of maintaining healthy eating habits, such as consuming a variety of fruits and vegetables, in reducing the risk of developing health issues like metabolic syndrome (MS) and other chronic diseases. This emphasizes the role of diet in overall health and well-being. Further, the third lady in discussion three expanded on this notion, saying,

*"I believe that adopting improved eating habits, such as reducing excessively salty meals, can act as a preventive step in decreasing the incidence of chronic diseases like MS among people living with HIV." (W3_FGD3) "*

The third participant's emphasis on reducing excessively salty meals as a preventive measure for chronic diseases like MS among people living with HIV indicates the importance of personalized dietary recommendations for individuals with specific health conditions. This suggests that diet modifications tailored to the needs of PLWHs can play a role in managing their health.

## 3.17. Regular physical activity performance as a positive practice

The study also found that "regular physical activity and spending more time indoors working may potentially lessen the risk of MS among PLWHs. This is demonstrated by the following examples of replies from respondents:

*"To ward off any illness, everyone should regularly engage in physical activity...Avoiding idleness is also important. You understand why I said this, right? Inactivity makes our legs constrict, which exposes us to other diseases and over time puts us at risk for developing a psychiatric disorder due to the additional stresses this health issue brings along with HIV/AIDS...Therefore, I believe that physical activity is necessary for everyone who has this viral infection."(MP_Ind2)*

Respondents emphasized the importance of physical activity and avoiding idleness, as inactivity can lead to various diseases and increased risk of developing psychiatric disorders. They believe that regular exercise is necessary for everyone with this viral infection, as inactivity can result in various diseases and an increased risk of psychological problems due to the added stress associated with HIV/AIDS. By promoting regular physical activity and avoiding idleness, there is potential to lower the risk of metabolic syndrome and other health complications. This underscores the role of exercise in managing health outcomes in this population.

Echoing this situation, the third woman in the first discussion reacted,

*"I must confess that I have truly benefited from dedicating significant time to indoor activities such as work, cooking, and attending to my responsibilities. For instance, I find greater value in these pursuits (W3_FGD1)."*

The third woman in the first discussion expressed her significant benefit from dedicating time to indoor activities like work, cooking, and responsibilities. She found greater value in these pursuits and found them to be beneficial. This highlights the value of engaging in productive tasks and responsibilities, which can contribute to overall well-being.

**3.17.1. Existential practice.** Airhihenbuwa describes Existential practices are those defined under the sociocultural perspectives that are not primarily unwanted or beneficial in their application. It put behaviors in a fresh context that had not before been well-thought-out [21, 33–35]. This study overall suggests that traditional practices like consuming 'Kocho,' 'Bulla,' 'Injera,' and 'Wot', along with beverages similar to Coca-Cola called 'kiineettoo' and religious beliefs, can serve as preventive measures for chronic health issues such as metabolic syndrome (MS) and its long-term complications in adults with HIV, despite the absence of scientific backing.

## 3.18. The practice of consuming locally noble foods as an existential practice

According to participant comments, the study contends that existential eating patterns associated with locally noble foods may be used as risk factor preventive techniques.

*"I believe our dietary approach... If feasible, consuming baked 'Koochoo' and porridge made from 'Bulla' (known locally as 'Bulla-Atimit') is advisable... Unless... the medication (ART drugs) we use to treat the illness (HIV) poisons and harms us..." (W5_FGD1)*

Participants believe that consuming baked 'Koochoo' and porridge made from 'Bulla' is advisable. These items are taken from the root portion of the 'Enset' plant, an organic, zonally endemic, indoor-cooked, and culturally acceptable food product with no scientifically proven health dangers. Participants in the study believe that consuming baked 'Koochoo' and porridge made from 'Bulla' can potentially mitigate the negative impact of HIV medication on their health. This indicates a practical and culturally grounded approach to managing the side effects of treatment. The participants' belief in the benefits of consuming specific local foods

also underscores the importance of cultural considerations in health management for individuals living with HIV. By incorporating traditional noble foods into their diet, participants are utilizing cultural knowledge and practices to potentially improve their health outcomes and reduce the risk of metabolic syndrome. This highlights the value of culturally relevant interventions in promoting well-being. The third woman in the initial debate responded in line with this circumstance, saying,

> *"Although eating 'Injera' on a daily basis has health benefits. Using 'Shiro-Wot' in particular in conjunction with it will yield better outcomes." (FGD4_M1)*

She said that the daily consumption of Ethiopian staple food 'Injera' can improve health outcomes by combining it with 'Shiro-Wot', an indoor stew made from processed pea, bean, and chickpea grains. Injera, made from the grain 'Teff', is known for its rich nutritional profile, including fiber and various nutrients. Combining 'Injera' with 'Shiro-Wot' provides a balanced and nutritious meal that provides essential nutrients for overall health and well-being. These foods are deeply rooted in Ethiopian cultural heritage and dietary traditions, and are often enjoyed as part of everyday meals or special occasions. The findings suggest that combining 'Injera' and 'Shiro-Wot' may help prevent metabolic syndrome among HIV-positive adults. While the scientific relevance of this claim is not yet proven, the cultural and dietary significance of these foods in Ethiopian cuisine may offer unique benefits that contribute to overall health and well-being. Further scientific investigation is needed to validate their efficacy and explore the impact of these traditional Ethiopian foods on metabolic health outcomes and determine their role in a holistic approach to health promotion for individuals living with HIV.

### 3.19. "Kiineettoo" in Amharic, or "Coca-Cola-like beverages," as an existential practice

Additionally, the study analysis suggests that consuming beverages similar to Coca-Cola, referred to as "kiineettoo" in Amharic, may assist in preventing metabolic syndrome (MS) in HIV-positive adults (PLWHs) based on respondent comments.

> *"Given that we live with the virus. . . we avoid alcoholic beverages that can intoxicate us," another participant mentioned. However, we occasionally consume 'kiineettoo' (a locally produced Coca-Cola-like drink)" (W1_FGD3).*

As noted in the prior comment, drinking Coca-Cola-like beverages, called "kiineettoo" in Amharic, might aid in averting metabolic syndrome among HIV-positive adults. The information provided suggests that PLWHs, who typically abstain from alcohol due to its potential negative interactions with medication or health concerns, may occasionally consume a locally produced fermented drink called "kiineettoo" as an alternative. This drink, which tastes like Coca-Cola, is referred to as existential drinking in Amharic. "Kiineettoo" is a favored existential beverage for HIV-positive persons, mirroring cultural and societal standards. By opting for this indigenous substitute, people can engage in social events without endangering their health. It serves as a symbolic link to their cultural legacy and customs, underlining a deliberate focus on health and welfare during social interactions. The wide embrace of "kiineettoo" further fosters community approval and inclusiveness, enabling participation in shared events and social meets.Religious beliefs as existential practice:

### 3.20. Religious beliefs as existential practice

In addition, as noted by multiple respondents, people with HIV view their relationship with God as an existential practice. The quote below illustrates this idea, showing how their religious beliefs influence their actions and choices regarding health.

*"Like most communities, including mine, are religious. . . um. . . This means that when we live in accordance with our faith, we are shielded from various health issues, even if we don't currently have any. Simply by following our faith; furthermore, we are able to avoid unnecessary behaviors such as alcoholism, smoking, and infidelity; we encounter no difficulties"* *(W3_FGD3).*

This implies that recognizing religious beliefs as a key aspect of avoiding risks and improving health for those with HIV underscores the need to address spiritual and existential well-being in healthcare support. Including culturally appropriate and faith-centered methods in their treatment strategies could improve their health results and well-being.

**3.20.1. Negative practices.**   Airhihenbuwa defines negative practices as behaviors rooted in undesirable principles that result in adverse health outcomes. There is often a tendency to avoid self-blame when reflecting on these negative practices, which may lead to reluctance among individuals to seek or share information about them. The responses provided by participants throughout the discussion reflect this mindset, indicating that such practices can manifest as harmful or destructive behaviors [21, 33–35].The study specifically identified unhealthy eating habits, sedentary lifestyles, and substance abuse as negative behaviors that can contribute to the development of metabolic syndrome (MS) in adults living with HIV (PLWHs). Notable concerning practices included the consumption of raw meat, insufficient physical activity, and harmful substance use behaviors, such as excessive alcohol consumption, smoking, and drug use, all of which negatively impact the health of individuals with HIV. Participants underscored the detrimental effects of these behaviors on their overall well-being and discussed the challenges they face in managing these habits.

### 3.21. Unhealthy feeding habits as negative practices

The data set revealed that the consumption of raw meat, a culturally preferred food in the community, is a negative eating habit that worsens the progression of MS risks and its long-term effects. Most respondents in the study addressed this issue in their comments. One subject remarked,

*"I perceive the prevalent habit of frequently consuming 'raw meat' (known as 'Tire-Sigga' in Amharic) here as one of the bad practices associated with our eating habits." (M8_FGD2)*

Another participant stated,

*"I believe we shouldn't consume fatty meat and fatty food products because these products cause diarrhea and other difficulties in those infected with the virus." (W1_FGD1)*

The above extracts revealed that the consumption of raw meat, a culturally preferred food, is a negative eating habit that worsens metabolic syndrome risks and long-term effects among adults living with HIV. Most respondents viewed this habit as a bad practice, while one participant argued against consuming fatty meat and food products, as they cause diarrhea and other difficulties in those infected with the virus. The finding highlight the detrimental effects of

frequent consumption of raw beef and fatty foods on participants' health, despite their cultural beliefs and traditional eating habits, underscoring the importance of addressing these issues in healthy eating habits.

The study also explores the reasoning behind consuming 'Injera' with meat-Wot, a meat-based sauce, as a potential way to promote health and enhance strength, despite its potential health risks. For instance, a participant mentioned,

*"Our practice of eating 'Injera' with 'meat-Wot,' a sauce made with meat as an ingredient (known as 'Sigga-Wot' in Amharic) is a sensible choice. Such dietary choices can be beneficial for individuals like us (living with HIV infection) to avoid health issues and improve strength." (W3_FGD3)*

A participant believes this dietary choice can help individuals like them living with HIV avoid health issues and improve strength, despite the potential health risks. The study highlights the importance of considering these dietary choices when preparing meals. Overall, these findings underscore the complexity of addressing dietary practices and cultural influences in managing metabolic syndrome risks among adults living with HIV. Healthcare providers and support services should take into account cultural beliefs and preferences when developing interventions to promote healthier dietary choices and improve overall health outcomes for this population.

## 3.22. Sedentary lifestyles as negative practices

Participants acknowledged that unhealthy habits often stem from being sedentary. Their statements supported this notion. For instance, one participant disclosed,

*"Being infected with HIV doesn't afford you time but rather leaves you vulnerable to various illnesses, hindering your ability to be active. It's as if you're confined to bed." (M3_FGD4)*

The interviews shed light on how HIV impacts People Living with HIV (PLWHs), resulting in deteriorating health, immobility, and sedentary behavior. They stressed how HIV infections and the challenges of living with the virus prompt these behaviors, which may worsen over time. This underscores the crucial need for improved support and care for PLWHs, as the virus exposes them to additional health risks and exacerbates existing issues.

Another participant remarked,

*"It's hard to even think, let alone act! Balancing our HIV condition with financial difficulties and the draining effect of daily medication creates a cycle that may influence our actions." (W7 FGD3)*

The speakers discussed the struggles faced by those managing HIV and financial burdens due to weakened capabilities from the virus and the daily medication regimen. They speculated that these factors could contribute to repetitive behaviors, impeding independent thinking. Moreover, participants highlighted the impact of unemployment, leading to idle time and increased health risks, such as potential risks like MS. The study highlights the harmful effects of HIV on PLWHs. Participants note a negative view of a sedentary lifestyle due to its link to unhealthy behaviors. Feedback emphasizes HIV's impact, resulting in poor health, extended bed rest, and reduced physical activity. Participants discuss how living with HIV worsens these issues, requiring improved support and treatment. Financial difficulties and daily medication use add to the challenges for those with HIV. Unemployment is highlighted as a factor

increasing health risks, supporting the study's conclusions on HIV's adverse effects on individual well-being.

### 3.23. Harmful substance use as negative practice

The analysis indicated that engaging in harmful substance use practices among individuals with HIV could exacerbate the occurrence of metabolic syndrome and related risks. For instance, one study participant highlighted how young people, particularly the most addicted, often spend their earnings on consuming a locally produced alcohol known as 'Areke,' despite financial constraints.

*"The youth, being the most addicted, often use their daily labor wages to drink a locally prepared alcohol called 'Areke' to get drunk" (M2_FGD4).*

In contrast, another participant expressed the belief that individuals with HIV should refrain from alcohol consumption to avoid potential difficulties stemming from lack of self-regulation.

*"In my perspective, I believe we should abstain from consuming alcohol. . .Because we can't regulate ourselves, we'll be exposed to a slew of difficulties" (M7_FGD2).*

Additionally, a different perspective was shared by a participant who mentioned using alcohol and cigarettes as coping mechanisms for the stress and agitation associated with living with HIV, even though health experts advise against it.

*"As is well known, the major issue of living with HIV is. . .Because of the condition, you are frequently agitated. . .As a result, we may occasionally smoke a cigarette or consume alcohol to distract ourselves from the inconvenience. Health specialists frequently tell us that it creates another health issue. . .In contrast, we continue to use it covertly. . .by rejecting their recommendations. . .We should probably stop our harmful habit."(W3_FGD3).*

These discussions underscore the complexities surrounding alcohol use, HIV infection, and health outcomes. This situation emphasizes the importance of addressing the long-term impacts of alcohol on individuals living with HIV, including psychological challenges and metabolic syndrome. Furthermore, it highlights the necessity of addressing the psychological issues linked to alcohol use in this context. Furthermore, as evidenced in the response below, the perception surrounding smoking has shifted concerning risky substance abuse habits.

*"People residing in this area frequently encounter unnecessary addictions like. . .smoking cigarettes. . ." (M7_FGD2)*

As per the provided response, the narrative related to smoking has progressed, with residents of the area being subjected to negative practices like cigarette smoking. It was commonly noted that smoking cigarettes is a detrimental substance with adverse effects on one's well-being. One individual expressed:

*"As it is widely recognized, a key issue when living with HIV is. . .constant frustration due to the condition. . .Therefore, we might indulge in a cigarette from time to time. . .Despite the repeated warnings from healthcare professionals about the risks it poses to our health, we*

*persist in our clandestine use. . .by disregarding their advice. . .It's probably time to break away from our unhealthy patterns. . ."(W3_FGD3)*

The speaker highlighted that individuals with HIV often resort to smoking covertly due to temporary anger relief, despite healthcare providers' counsel. Dealing with HIV-induced irritation may drive individuals to smoke occasionally. Despite the frequent warnings, many individuals persist in smoking, neglecting the importance of abandoning harmful practices. Another statement indicates shifts in the environment surrounding the consumption of "Khat" in relation to substance use.

*"People commonly spend their days idly, often chewing 'Khat'. . .It's a usual occurrence. . .Even among those I know who have the virus and frequently partake in this stimulant, "I suspect it could be detrimental" (M2_FGD2).*

Conversely, an individual in the fourth discussion remarked,

*"By initiating change and exerting effort. . .For instance, we need to. . .refrain from negative habits, such as smoking, 'Khat,' and alcohol" (M1_FGD4).*

Participants expressed worries about the consequences of inaction, particularly focusing on the use of 'Khat,' a stimulant commonly used by those with HIV. In essence, the research uncovers harmful substance abuse patterns, notably alcohol, smoking, and 'Khat' consumption, associated with adverse behaviors, particularly among youths and individuals living with HIV, necessitating immediate intervention.

## 4. Discussion

This study explores the sociocultural factors that affect the understanding of metabolic syndrome (MS) among adults living with HIV in Ethiopia's Southern Regional state, using Airhihenbuwa's PEN-3 model and the Knowledge, Attitudes, and Practices (KAP) framework. The findings reveal a significant knowledge gap regarding MS and its risk factors, leading to poor health attitudes and practices among people living with HIV. Sociocultural elements, such as beliefs, values, family dynamics, and community support, are crucial in shaping these individuals' knowledge and approaches to preventing and managing chronic diseases like MS.

Airhihenbuwa's PEN-3 model provides a nuanced framework for understanding how cultural factors affect health knowledge and behaviors, especially among adults living with HIV. By emphasizing the interconnected areas of Individual, Extended Family, and Neighborhood, the model highlights the importance of both personal experiences and community dynamics in shaping health-related knowledge and behaviors [32, 34, 54–56]. The study indicates that PLWHs access health information from a range of sources, including personal experiences, mass media, and peer discussions. This suggests a rich tapestry of information dissemination that tailored health education strategies can tap into. By utilizing these channels, public health initiatives can enhance the awareness and understanding of metabolic syndrome risks and prevention strategies among this population. The emphasis on cultural beliefs, personal experiences, and the role of mass media underscores the need for health education efforts to be culturally sensitive and contextually relevant.

The research points out that extended family member serve as vital sources of health information for individuals living with HIV. This finding illuminates how familial support systems can be leveraged to enhance knowledge about metabolic syndrome and its prevention. Given the observed gaps in awareness, it suggests that educational efforts should not only target

PLWHs but also their families to equip them with the knowledge necessary to support healthier lifestyle choices. Initiatives that involve family members in educational campaigns could foster a more supportive environment for behavioral changes among PLWHs.

The identification of both formal and informal channels, such as religious organizations and social groups, as effective in health education activities underscores the potential for community engagement in health promotion. Collaborative efforts with community-based organizations can help amplify the reach of health education initiatives while ensuring that they are culturally sensitive and relevant. Such partnerships can also foster community ownership of health education campaigns, enhancing their legitimacy and effectiveness.

According to the PEN-3 model [32, 34, 54–56], an individual's connection to family and neighborhood customs is crucial for acculturation and health education. Importantly, participant feedback revealed significant gaps in knowledge regarding metabolic syndrome, its associated risks, and preventive measures. The study highlights a need for educational strategies that take a holistic view of chronic illnesses and their interrelatedness, such as high blood pressure and diabetes. Developing resources that connect chronic illnesses with lifestyle factors can empower PLWHs to take proactive steps towards health management. Moreover, enhancing health literacy within this population can facilitate informed decision-making and promote self-efficacy in managing their health.

Another vital aspect of the study is its emphasis on viewing metabolic syndrome within the larger framework of chronic illness management. By encouraging PLWHs to consider the interconnections between various health conditions, the research advocates for a holistic approach to health care. This perspective can lead to better strategies that encompass not only diet and exercise but also mental health and social well-being. The findings reinforce the necessity for culturally sensitive interventions that are considerate of the diverse backgrounds within the HIV-positive population. Health educators and healthcare providers must invest in understanding cultural nuances that influence health behaviors and knowledge. Such sensitivity will enhance the effectiveness of health interventions aimed at reducing the risks of metabolic syndrome and improving overall health outcomes for PLWHs.

In general, the study's findings make a compelling case for the integration of sociocultural factors into health education initiatives tailored to individuals living with HIV. By leveraging Airhihenbuwa's PEN-3 model alongside community dynamics, targeted educational efforts can significantly enhance knowledge about metabolic syndrome, encourage healthier lifestyle choices, and ultimately improve the well-being of PLWHs. Future research and intervention programs should continue to emphasize cultural empowerment, familial involvement, and community engagement as essential components of effective health promotion strategies.

The application of the PEN-3 model to investigate health behaviors among adults living with HIV (PLWHs) reveals critical insights into how sociocultural factors fundamentally shape perceptions, beliefs, and attitudes toward metabolic syndrome (MS). By examining relationship and expectation (RE) domains, we can begin to unpack the complexities of health behavior within this population, particularly their understanding and management of coexisting health risks like MS [32, 34, 54–56]. This study applied these concepts to investigate how the knowledge, beliefs, and values surrounding the risks and prevention of metabolic syndrome (MS) held and communicated by the participants, their families, and communities impact their engagement with the subject.

The study finding unveiled discrepancies in the perspectives of adults living with HIV (PLWHs) concerning the risks associated with metabolic syndrome (MS). The discrepancies in how PLWHs view MS as a health issue compared to HIV highlight a significant challenge in health communication and education. Most participants perceive HIV as a far more significant threat, thereby diminishing the urgency associated with managing other health risks such as

MS. This scenario underscores the importance of addressing underlying beliefs and values that shape health behaviors. If PLWHs are not well-informed about the potential risks of MS, their health management strategies may remain inadequate. Moreover, it's crucial to recognize that health attitudes are not formed in isolation; they are influenced by the social circles surrounding individuals. The study confirms that family members, peers, and community networks play pivotal roles in shaping these attitudes. Therefore, health promotion efforts must recognize and engage these social dynamics to ensure that messages about the importance of managing metabolic syndrome are effectively disseminated and internalized.

The enabler and nurturer domains of the PEN-3 model emphasize the role of societal and cultural factors that impact health-related behavior changes [32, 34, 54–56]. The findings mentioned in your discussion highlight that PLWHs' responses to health information about MS are influenced by several interconnected factors, such as the perceived severity of HIV, economic circumstances, and cultural norms. For example, family support can serve as a significant facilitator in adopting healthier lifestyle choices. In contrast, stigma associated with both HIV and MS can hinder open discussions about these health issues, leading to further misinformation and negative health outcomes. Therefore, interventions should not only aim to educate PLWHs about MS but also to empower their support networks to foster environments where healthy behaviors are encouraged and normalized.

The research underscores the pivotal role caregivers can play in influencing PLWHs' perspectives regarding MS. Caregivers often have the most direct influence over health behaviors, providing both emotional support and practical guidance. This finding suggests that training and equipping caregivers with the knowledge and tools to promote healthy lifestyles can significantly improve health outcomes. Involving caregivers in educational initiatives tailored to address both HIV and MS can help create a more cohesive understanding of health management. The necessity for collaboration between government and non-governmental organizations (NGOs) to tackle public health challenges like HIV and MS cannot be overstated. By promoting healthy lifestyle choices and providing comprehensive support structures, these collective efforts can effectively mitigate the public health threats posed by both conditions. Strategies could include community health initiatives, outreach programs, and partnerships with local organizations to enhance access to resources, information, and support.

The study also points out the importance of tackling the sociocultural challenges that influence the attitudes of PLWHs toward MS. Public health initiatives focused on reducing stigma through education and awareness campaigns are vital to reshaping societal perceptions and encouraging compassionate discussions around health management. By fostering an inclusive environment, we can diminish the negative perceptions associated with both HIV and MS, empowering individuals to take proactive steps in managing their health. Overall, applying the PEN-3 model to understand the health behaviors of PLWHs concerning metabolic syndrome enables a comprehensive analysis of the multifaceted influences on health perceptions and behaviors. Emphasizing the importance of social networks, cultural norms, and support systems can lead to more effective health interventions. As we continue to explore these dynamics, strategic initiatives focusing on education, stigma reduction, and enhanced support services can create a more favorable environment for PLWHs to manage metabolic syndrome and improve their overall health outcomes. Ultimately, a concerted effort toward addressing these issues will yield healthier communities, greater awareness, and more compassionate responses to the complex interplay of chronic health conditions among individuals living with HIV.

Airhihenbuwa's model provides a rigorous examination of the role of sociocultural factors in shaping health behaviors among individuals living with HIV, specifically through the lens of Cultural Empowerment (CE) within the PEN-3 model. This model offers a comprehensive

framework by which to understand the interactions between culture, health behavior, and individual empowerment. By emphasizing the importance of positive practices and the move from self-blame to constructive perspectives, the study highlights how a culturally attuned approach can profoundly influence health outcomes [32, 34, 54–56]. The study identification of positive habits—such as consuming a diverse range of fruits and vegetables and engaging in regular physical activity—illustrates a proactive approach to health management that can significantly reduce the risk of metabolic syndrome (MS) among PLWHs. The emphasis on healthy eating not only addresses physical health but also serves as a form of cultural expression and community solidarity. This finding aligns with broader public health principles that advocate for lifestyle modifications as an integral part of managing chronic health conditions.

Furthermore, the research's exploration of existential practices, including traditional dietary habits and religious beliefs, provides critical insight into how deeply embedded cultural norms can offer protective factors against health risks. By recognizing traditional foods—like "Kocho," "Bulla," "Injera," and "Wot"—as potential health enhancers, the study advocates for a deeper integration of cultural wisdom into healthcare interventions. This aligns with the growing acknowledgment of the significance of culturally relevant healthcare practices that respect and utilize patients' backgrounds and beliefs to foster trust and engage communities. Conversely, the identification of negative behaviors associated with metabolic syndrome presents a stark reminder of the complexities involved in health promotion within vulnerable populations like those living with HIV. The study points out behaviors that exacerbate health risks, such as poor dietary choices, high rates of substance abuse, and sedentary lifestyles, which are often rooted in socioeconomic factors and systemic barriers. This highlights an urgent need for educational and support initiatives tailored to address these detrimental behaviors, particularly in settings where individuals may lack access to resources or face stigma.

The findings underscore the importance of culturally sensitive interventions that consider the unique sociocultural contexts of individuals living with HIV, particularly in regions like Ethiopia where cultural diversity is pronounced. Tailored educational initiatives that respect and incorporate cultural practices can foster better understanding and adoption of healthy behaviors. By integrating culturally relevant strategies into public health campaigns, health professionals can promote meaningful engagement and behavior change.

Additionally, the study revealed significant challenges posed by inadequate knowledge, negative attitudes, and ineffective practices related to Metabolic Syndrome (MS) and its lifestyle risk reduction strategies among individuals living with HIV. It pointed to marked disparities in health behaviors, particularly concerning diet, physical activity, and substance use in managing MS. This finding requires targeted educational interventions. The study highlights that despite some awareness of the risks associated with MS, there are marked disparities in knowledge about lifestyle choices that can mitigate these risks. Tailoring health communication to address these knowledge gaps, while transforming negative perceptions into constructive discussions about health, is essential for effective management. This study advocates for a comprehensive approach that intertwines positive, existential, and negative practices within a culturally sensitive framework. Healthcare providers need to account for the interactions among these elements when designing health promotion strategies aimed at empowering individuals living with HIV. Establishing collaborations with community organizations, utilizing local insights, and involving patients as active participants in their healthcare can enhance the management of both HIV and metabolic syndrome. By weaving sociocultural factors into health promotion strategies, the research outlines avenues for addressing knowledge deficits, transforming negative perceptions, and fostering healthier behaviors—essential elements for reducing the risks associated with metabolic syndrome, especially within the African context, and particularly in Ethiopia. This holistic approach emphasizes the importance of harnessing

cultural contexts to provide effective health management for individuals. In conclusion, Airhihenbuwa's research not only pinpoints vital pathways for encouraging positive health behaviors among HIV-positive individuals but also underscores the significance of cultural empowerment in improving overall health. To navigate the intricate landscape of sociocultural influences on health behaviors, developing inclusive and effective health interventions that resonate with the targeted populations is crucial. By prioritizing culturally relevant practices, we can create environments that both honor and enhance the informed choices of individuals living with HIV, ultimately leading to improved health outcomes and quality of life.

## 5. Conclusion

This study reveals a critical gap in knowledge regarding metabolic syndrome (MS) and its risk factors among adults people living with HIV/AIDS (PLWHs), contributing to adverse health attitudes and suboptimal health practices within this population. Although some awareness exists about the risks associated with MS, significant deficiencies persist in preventive practices. Central to these findings is the importance of"sociocultural contexts"—including beliefs, values, family dynamics, and community support—shaping the knowledge, attitudes, and behaviors of PLWHs concerning chronic disease prevention and management. The effectiveness of health education initiatives is heavily dependent on incorporating formal and informal support networks that can influence positive behavior changes. The study also emphasizes the value of diverse information sources, such as personal experiences, mass media, and peer discussions, in disseminating crucial health-related knowledge to PLWHs. To improve health outcomes, the research advocates for culturally sensitive interventions that are attuned to the distinctive contexts and needs of this population. Recommended practices for preventing MS among PLWHs include promoting a balanced diet rich in fruits and vegetables, reducing salt intake, encouraging regular physical activity, and addressing harmful behaviors such as unhealthy dietary choices and substance abuse. Empowering individuals through targeted education programs and culturally relevant health campaigns is essential for fostering healthier lifestyle choices and bridging significant knowledge gaps. Future research should examine local health practices and their efficacy among PLWHs in culturally specific settings, particularly in areas like the Gedeo zone in Ethiopia, to gather scientific data that supports effective community-driven interventions. Additionally, establishing comprehensive support networks that provide psychological, social, and medical assistance could significantly improve the overall well-being of HIV-positive individuals in African in general and in Ethiopia in particular. and facilitate behavior change.

## Supporting information

**S1 Fig. Adapted study's conceptual framework.**
(DOCX)

**S1 File. The SRQR reporting guidelines completed checklist.**
(PDF)

**S1 Table. Additional quotes from participants.**
(XLSX)

**S2 Table. A Code book used as a guide for analysis.**
(XLSX)

**S3 Table. Connection of the research question.**
(DOCX)

## Acknowledgments

My gratitude goes to Addis Abeba University and Dilla University for their financial support. I would also like to express my gratitude to the staffs of Dilla University College of Medicine and Health Science, Wonago district health office, Dilla University Referral Hospital, and Wonago health center, as well as their health professionals, particularly those working in HIV clinics, for their unqualified assistance with data collection. Furthermore, I would want to express my profound appreciation to all research participants for their willingness to engage in this experiment.

## Author Contributions

**Conceptualization:** Girma Tenkolu Bune.

**Data curation:** Girma Tenkolu Bune.

**Formal analysis:** Girma Tenkolu Bune.

**Funding acquisition:** Girma Tenkolu Bune.

**Investigation:** Girma Tenkolu Bune.

**Methodology:** Girma Tenkolu Bune.

**Project administration:** Girma Tenkolu Bune.

**Resources:** Girma Tenkolu Bune.

**Software:** Girma Tenkolu Bune.

**Supervision:** Girma Tenkolu Bune.

**Visualization:** Girma Tenkolu Bune.

**Writing – original draft:** Girma Tenkolu Bune.

**Writing – review & editing:** Girma Tenkolu Bune.

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
