## [Decision Letter · Decision Letter 0]

17 May 2024

PONE-D-24-06838The Impact of Sociocultural Contexts on the Knowledge, Attitudes, and Practices of Adults Living with HIV/AIDS in Ethiopia towards Metabolic Syndrome Risks:A Descriptive Phenomenology Study Using the PEN-3 ModelPLOS ONE

Dear Dr. Bune,

Thank you for submitting your manuscript to PLOS ONE. After careful consideration, we feel that it has merit but does not fully meet PLOS ONE’s publication criteria as it currently stands. Therefore, we invite you to submit a revised version of the manuscript that addresses the points raised during the review process. Please submit your revised manuscript by Jul 01 2024 11:59PM. If you will need more time than this to complete your revisions, please reply to this message or contact the journal office at plosone@plos.org. Please include the following items when submitting your revised manuscript:A rebuttal letter that responds to each point raised by the academic editor and reviewer(s). You should upload this letter as a separate file labeled 'Response to Reviewers'.A marked-up copy of your manuscript that highlights changes made to the original version. You should upload this as a separate file labeled 'Revised Manuscript with Track Changes'.An unmarked version of your revised paper without tracked changes. You should upload this as a separate file labeled 'Manuscript'.

We look forward to receiving your revised manuscript.

Kind regards,

Adedayo Ajidahun

Academic Editor

PLOS ONE

Journal Requirements:

2. Please provide additional details regarding participant consent. In the ethics statement in the Methods and online submission information, please ensure that you have specified (1) whether consent was informed and (2) what type you obtained (for instance, written or verbal, and if verbal, how it was documented and witnessed).

Reviewers' comments:

Reviewer's Responses to Questions

**Comments to the Author**

1. Is the manuscript technically sound, and do the data support the conclusions?

Reviewer #1: Partly

Reviewer #2: Yes

2. Has the statistical analysis been performed appropriately and rigorously? 

Reviewer #1: N/A

Reviewer #2: N/A

3. Have the authors made all data underlying the findings in their manuscript fully available?

Reviewer #1: Yes

Reviewer #2: No

4. Is the manuscript presented in an intelligible fashion and written in standard English?

Reviewer #1: Yes

Reviewer #2: Yes

5. Review Comments to the Author

Reviewer #1: The manuscript focuses on a pertinent issue in HIV care: PLWH knowledge, attitudes and practices (KAP) about Metabolic Syndrome.

Introduction section: lacked in-depth literature review on previous studies that examined the same topic and studies that used the PEN-3 model, and why the study was conducted. For example, are there other studies that have examined same topic among PLWH in other countries, or even in Ethiopia? What were the findings? What gaps and in what way does this study address them? Why was the PEN-3 model selected as the analytical tool? Specifically, besides the PEN-3 model being developed for African contexts, in what ways (as shown in other studies) is it the appropriate model? Also, there is the inconsistent use of the acronym: PLWH or PLWHA.

Methods section: fragmented in its current state. It should be organized for clarity and flow. Since the focus of the study is to examine sociocultural context, more information is needed about the study setting – location, languages spoken, etc. For example, Why Gedeo zone of Ethiopia? What makes that zone the location for the study? It is unclear what language the FG discussions were conducted in: English or local language? While there is reference in the result section about the use of local words to describe some things, it’s unclear whether this was the language used in data collection. Was the discussion questions in English or translated into the language of the community? This is necessary if the goal is to explore the sociocultural context as language is the conduit through which thoughts, feelings and perspectives are expressed, and it’s a product of the sociocultural context of where is it used. Also, there is the use of acronym not explained: DURH, WHC? On page 7, line 8: what was pretested? It’s not clearly stated. Though the author(s) included a table showing how the PEN-3 model was used, there is lack of depth in how the model was actually used. For example, was the PEN-3 model used in the development of FGD questions? Was it used for analysis of the data after the study was conducted? It seems overly simplified to use the PEN-3 model to aggregates items in the domains without thorough engagement of the tool-kit. In its current state of the data presentation, it conveys the idea that the model was used after the study was conducted to guide data “arrangement” not necessarily data analysis. That is, the PEN-3 model was used in organizing the data, and not used in guiding the data analysis. If this is the case, then it should be clearly stated.

Data Analysis: How was the codebook generated? (page 9, line 3-4). As stated above, in its current state, the PEN-3 model was used to arrange the data not data analysis.

Results section: This needs to be seriously re-organized to focus on presenting the results of the study. There is a mix-up with comments from the author and participants’ quotes making for lack of clarity. Participants’ quotes to support the points or themes should be presented clearly. Also, there is confusing focus on the purpose of the study based on the comments/quotes of participants. Is the focus on Metabolic Syndrome or Multiple Sclerosis? The author needs to define the study focus, and should indicate if participants introduced a topic and/or idea “outside” of the focus of the study, like Multiple Sclerosis. So, it is unclear from the study what the sociocultural contexts of metabolic syndrome and NCDs are in the current state of the manuscript. The use of words like “individuals living with HIV lack sufficient contextual knowledge and value system” (pg. 22) belies the purpose of the study. People’s value system is product of their sociocultural context and serves as a lens in interpreting their experiences (phenomenon).

Discussion section: like the introduction, this section also lacks depth though the author included many references as to the use of the PEN-3 model. Also, the lack of clarity in the results section, and the use of phrases that belies the purpose of the study, makes the discussion and conclusion section weak in achieving the goal of the study: sociocultural factors on KAP of PLWH. In its current state, it did not show that the PEN-3 model was the theoretical basis for the study, neither did the study show “that sociocultural contexts have a significant role in the lack of awareness regarding risk factors and prevention methods for MS among PLWHs.

Reviewer #2: This study provides significant results that can guide future interventions in Ethiopia and In Africa in general.

Here are some minor revisions needed

- The abstract should introduce the PEN-3 model

- line 10 'Ethiopian studies' could be changed to ' studies in Ethiopia

- The discussion should include a paragraph on how cultural identity has been shown to influence health behaviors in the Ethiopian context in both positive and negative ways

- It should also include how cultural empowerment plays in the African context in general and in Ethiopia in particular.

6. PLOS authors have the option to publish the peer review history of their article (what does this mean?). If published, this will include your full peer review and any attached files.

Reviewer #1: **Yes: **Titilayo A Okoror

Reviewer #2: No

---

## [Author Response · Author response to Decision Letter 0]

22 Jul 2024

III. Section 3: Reviewer#1: Comment to the authors and Responses given by the authors 

Please use the space provided to explain your answers to the questions above. You may also include additional comments for the author, including concerns about dual publication, research ethics, or publication ethics. (Please upload your review as an attachment if it exceeds 20,000 characters). 

1. Introduction Section:

a) Reviewer#1: Comment to the authors: 

The manuscript focuses on a pertinent issue in HIV care: PLWH knowledge, attitudes and practices (KAP) about Metabolic Syndrome. 

A. .Introduction section lacked in-depth literature review on previous studies that examined the same topic and studies that used the PEN-3 model, and why the study was conducted. For example, are there other studies that have examined same topic among PLWH in other countries, or even in Ethiopia? What were the findings? What gaps and in what way does this study address them? 

B. Why was the PEN-3 model selected as the analytical tool? Specifically, besides the PEN-3 model being developed for African contexts, in what ways (as shown in other studies) is it the appropriate model? 

C. Also, there is the inconsistent use of the acronym: PLWH or PLWHA.

Responses to Reviewer #1

Concerns Regarding the Introduction Section by the Authors

Dear Reviewer #1,

We appreciate your emphasis on the importance of conducting a thorough literature review in the Introduction section. Your suggestion to delve deeper into previous studies relevant to the study topic and utilize the PEN-3 model has been taken into consideration. Please find below the responses addressing the concerns raised in the Introduction sections listed as A, B, and C.

A. In response to the feedback provided regarding the “A” section of Introduction section of our manuscript, we have taken into consideration the suggestions and made revisions to address the perceived shortcomings. The study presented in our paper utilizes the PEN-3 model as a theoretical framework to explore the sociocultural factors associated with metabolic syndrome among people living with HIV (PLWH). 

To provide a more comprehensive overview of previous studies that have utilized the PEN-3 model, we have examined a selection of four research articles that have employed this theoretical foundation. These articles were chosen after a thorough review of the literature, which included the screening of a total of 13,705 global studies, with a focus on studies conducted in Africa, particularly in Sub-Saharan Africa and Ethiopia. 

Our revised Introduction now includes a detailed discussion on how the PEN-3 model has been applied in various global studies focusing on communicable and non-communicable health issues. We have highlighted examples where the model has been used to understand sociocultural contexts related to obesity, diabetes, HIV/AIDS, cancer, smoking, and dietary habits among marginalized populations. Additionally, we have emphasized the importance of the model in developing context-specific interventions to address health challenges effectively.

Furthermore, we have provided insights into the methodological approaches used in studies that have employed the PEN-3 model, including qualitative, quantitative, and mixed methods. Specifically, we have discussed how the model has been utilized to explore the intersection of cultural contexts with health issues, with a particular focus on HIV/AIDS and non-communicable diseases.

We have also expanded on previous research that has applied the PEN-3 model in Africa and Sub-Saharan Africa, emphasizing the diverse range of health outcomes that have been explored using this framework. Examples include studies on cancer screening, hypertension, HIV/AIDS self-testing, and diabetes management, among others.

In the context of Ethiopian studies, we have incorporated information on the number of articles focusing on cultural perspectives related to health issues, the methodological approaches used, and the topics that have been examined. Additionally, we have highlighted the gap in the literature concerning the integration of the PEN-3 model in studies related to metabolic syndrome among PLWHs in Ethiopia. Thank you for the feedback, which has helped us enhance the clarity and depth of our Introduction section. For more clarity, please see the introduction part of the revised manuscript on page ____3____ to ____6_ Sincerely,

B. With respect to the issues of "why was the PEN-3 model selected as the analytical tool?" Once again, we value the feedback from the initial reviewer and aim to respond to the raised concerns regarding its use in our study in the following manner:

The decision to employ the PEN-3 model in this research, as opposed to other cultural frameworks, is grounded in four primary reasons:

The PEN-3 model is specifically tailored for African contexts, offering a unique perspective to identify culturally significant factors. Introduced in 1989, this model is instrumental in exploring individuals' attitudes, knowledge, and behaviors within their cultural milieu, particularly pertinent for examining public health issues from a sociocultural standpoint. Previous research has successfully utilized the PEN-3 model to analyze public health behaviors, providing a framework that integrates sociocultural contexts and pertinent factors into interventions. Originating from African contexts, notably East Africa and Ethiopia, it provides insight into public health challenges, aiding in recognizing influential factors in resource-limited settings. This model encourages the exploration of positive, negative, and distinct viewpoints, promoting dialogue and collaboration between researchers and communities to identify shared and differing perspectives. Comprising three key domains - cultural identity, relationships and expectations, and cultural empowerment - the PEN-3 model offers a comprehensive lens to understand how cultural dynamics interplay with public health issues. By delving into these domains, researchers can gain a holistic comprehension of the impact of cultural factors on health-related beliefs and behaviors, facilitating the development of targeted interventions aligned with the population's needs.

The model accentuates the positive socio-cultural influences on health behaviors and outcomes. It underscores the necessity to acknowledge and integrate these elements into culturally sensitive interventions, recognizing the significance of socio-cultural values in health-related initiatives. Understanding socio-cultural influences is pivotal in effectively addressing health issues. The PEN-3 model has been widely employed in global research across various health domains, including cancer, diabetes, HIV/AIDS, smoking, dietary choices, and obesity among marginalized communities referred to as "Others," contributing to over 100 studies globally. Its applications have paved the way for investigating how sociocultural contexts impact health behavior perceptions and for tailoring interventions to these contexts.

The model serves as a useful tool for data collection, analysis, and interpretation. Numerous scholarly works leverage the PEN-3 model to steer qualitative data collection and analysis techniques like categorization, cross-tabulation, and recontextualization, enhancing the understanding of health behaviors and the integration of culturally relevant elements into interventions.

The model underscores the pivotal role of family and community in shaping behaviors. Families and communities play a central role in influencing individuals' healthcare experiences and decisions, particularly in diverse healthcare settings. Family support is crucial in healthcare, emphasizing the necessity for tailored interventions to enhance care and reduce stigma for individuals, such as those affected by HIV/AIDS. Within the PEN-3 framework, importance is placed on community perspectives through qualitative inquiries, aiming to uncover perceptions of metabolic issues, knowledge of risk factors, and attitudes and behaviors concerning prevention and control. In addition to its African roots, these rationales provide further insights into the preference for this model. We have endeavored to address these critiques in line with the reviewers' feedback. Please refer to the updated introduction section of the manuscript for additional details. For more clarity, please see the introduction part of the revised manuscript on page ____4____ to ____6_ starting from the third paragraphes upto the end.

C. Regarding the inconsistent use of the acronym, we recognize the importance of clarity and consistency in terminology. Following a review of the literature, we ensured uniformity in referring to adults living with HIV (PLWHs) throughout the manuscript.

1. Methods section: 

A. Fragmented in its current state. It should be organized for clarity and flow. Since the focus of the study is to examine sociocultural context, more information is needed about the study setting – location, languages spoken, etc. For example, Why Gedeo zone of Ethiopia? What makes that zone the location for the study?

B. It is unclear what language the FG discussions were conducted in: English or local language? While there is reference in the result section about the use of local words to describe some things, it’s unclear whether this was the language used in data collection. Was the discussion questions in English or translated into the language of the community? This is necessary if the goal is to explore the sociocultural context as language is the conduit through which thoughts, feelings and perspectives are expressed, and it’s a product of the sociocultural context of where it is used. 

C. Also, there is the use of acronym not explained: DURH, WHC? On page 7, line 8: what was pretested? It’s not clearly stated. 

D. Though the author(s) included a table showing how the PEN-3 model was used, there is lack of depth in how the model was actually used. For example, was the PEN-3 model used in the development of FGD questions? Was it used for analysis of the data after the study was conducted? 

E. It seems overly simplified to use the PEN-3 model to aggregates items in the domains without thorough engagement of the tool-kit. In its current state of the data presentation, it conveys the idea that the model was used after the study was conducted to guide data “arrangement” not necessarily data analysis. That is, the PEN-3 model was used in organizing the data, and not used in guiding the data analysis. If this is the case, then it should be clearly stated. Data Analysis: How was the codebook generated? (page 9, line 3-4). As stated above, in its current state, the PEN-3 model was used to arrange the data not data analysis.

Responses given to the Reviewer #1, concerns rose in the method section by the authors

Dear Reviewer #1,

We appreciate your emphasis on the importance of conducting a thorough Methods Section. In response to feedback, we enhanced clarity and organization of the Methods section in the manuscript, including details about the study setting in the Gedeo zone of Ethiopia. The revised section addresses questions raised by reviewers on why this location was chosen for the study. Please find below the responses addressing the concerns raised in the Introduction sections listed as A, B, C, D, and E. 

A. About Fragmented flow: The Method section now follows the journal's recommended presentation outline. Refer to the revised manuscript for a clear flow of ideas. For more clarity please see the revised manuscript method section on page ____6____ to the end. 

B. Regarding the Study Setting: The study took place in the Gedeo zone of Southern Ethiopia, chosen for its unique sociocultural characteristics relevant to researching the influence of sociocultural contexts on the behavior of people living with HIV/AIDS (PLWHs) towards metabolic syndrome (MS) and associated risks. The diverse population in the Gedeo zone, with various cultural practices and languages such as "Amaharic" and "Gedeoffa," may impact health behaviors among PLWHs. To address this, we specified in the Methods the language used during data collection, particularly in FG discussions conducted in the local working language of the Gedeo zone. Effective communication was ensured by including team members proficient in both "Amaharic and Gedeo language.” For more information, please refer to the method section, Sampling methods and procedures part in the subsection of Setting part on page 9 to 11. 

C. Concerning Unexplained Acronyms: In academic writing, clarity is key. Therefore, we already explained acronyms like DURH and WHC used in the manuscript. After reviewing the paper, we consistently referred to Dilla University Referral Hospitals (DURH) and Wonago Health Centers (WHC) throughout, ensuring readers' understanding by initially spelling out the full names followed by the acronyms in parentheses for subsequent mentions. For more information, please refer to the method section, Sampling methods and procedures part in the subsection of Setting of data collector part part on page 10, starting line No one to eight. 

D. 

E. About the PEN-3 model, it encompasses the cultural identity, relationship and expectation, and cultural empowerment domains. 

The model was utilized in both pre and post data collection phases. Prior to data collection, it guided the creation of Focus Group Discussion (FGD) questions. By integrating the PEN-3 model during question formulation, our aim was to pose culturally sensitive, contextually relevant questions in line with the study objectives. Following the data collection, the model was utilized to guide our analysis in three phases. In the initial phase, data capturing and transcription were conducted step by step, incorporating Dr. GirmaTenkolu (DGT) alone, while Sadat Mohamed (SM) manually used a pre-established codebook derived from a theoretical model and the Knowledge, Attitude, and Practice (KAP) conceptual frameworks. As mentioned earlier, the model constructs were employed for coding, categorizing, formulating subthemes and themes in this phase. It also facilitated the interpretation of word data and other data to uncover cultural nuances influencing participants' perspectives on metabolic syndrome risks. The model allowed for the exploration of positive, existential, and negative cultural factors that shape knowledge, attitudes, and practices in this context. Ultimately, its integration in question development and data analysis aimed for a comprehensive and culturally sensitive examination of sociocultural impacts on Ethiopian adults with HIV/AIDS. Its usage enhanced our findings, providing insights into the intricate interplay of culture, beliefs, and behaviors concerning metabolic syndrome risks. 

Concerning the statement, "Using the PEN-3 model to aggregate items in the domains without thoroughly engaging the tool-kit seems overly simplified.", we wish to clarify its central role in our study. Despite perceptions of prioritizing data organization over analysis, the model deeply influenced our research from formulating questions to interpreting results. Regarding data analysis, the model was integral, categorizing participant responses on metabolic syndrome risk perceptions into its positive, existential, and negative cultural factors. This categorization informed a comprehensive codebook, aiding in identifying data patterns and themes. Additionally, the model guided interpretation of focus group discussion findings, revealing cultural influences on participants' knowledge, attitudes, and practices related to metabolic syndrome risks. Our methodology section in the revised manuscript now explicitly outlines how the PEN-3 model guided both data organization and analysis processes, demonstrating its significant impact on shaping study outcomes. For a detailed insight, we encourage a review of the updated methodology section.

2. Results section: 

This needs to be seriously re-organized to focus on presenting the results of the study. 

A. There is a mix-up with comments from the author and participants’ quotes making for lack of clarity. Participants’ quotes to support the points or themes should be presented clearly. 

B. Also, there is confus

---

## [Decision Letter · Decision Letter 1]

1 Aug 2024

The Impact of Sociocultural Contexts on the Knowledge, Attitudes, and Practices of Adults Living with HIV/AIDS in Ethiopia towards Metabolic Syndrome Risks:A Descriptive Phenomenology Study Using the PEN-3 Model

PONE-D-24-06838R1

Dear Dr. Bune,

We’re pleased to inform you that your manuscript has been judged scientifically suitable for publication and will be formally accepted for publication once it meets all outstanding technical requirements.

Kind regards,

Adedayo Ajidahun

Academic Editor

PLOS ONE

Additional Editor Comments (optional):

Reviewers' comments:

Reviewer's Responses to Questions

**Comments to the Author**

1. If the authors have adequately addressed your comments raised in a previous round of review and you feel that this manuscript is now acceptable for publication, you may indicate that here to bypass the “Comments to the Author” section, enter your conflict of interest statement in the “Confidential to Editor” section, and submit your "Accept" recommendation.

Reviewer #2: All comments have been addressed

2. Is the manuscript technically sound, and do the data support the conclusions?

Reviewer #2: Yes

3. Has the statistical analysis been performed appropriately and rigorously? 

Reviewer #2: N/A

4. Have the authors made all data underlying the findings in their manuscript fully available?

Reviewer #2: Yes

5. Is the manuscript presented in an intelligible fashion and written in standard English?

Reviewer #2: (No Response)

6. Review Comments to the Author

Reviewer #2: (No Response)

7. PLOS authors have the option to publish the peer review history of their article (what does this mean?). If published, this will include your full peer review and any attached files.

Reviewer #2: No

---

## [Editor Report · Acceptance letter]

12 Aug 2024

PONE-D-24-06838R1 

PLOS ONE

Dear Dr. Bune, 

I'm pleased to inform you that your manuscript has been deemed suitable for publication in PLOS ONE. Congratulations! Your manuscript is now being handed over to our production team.

Kind regards, 

on behalf of

Dr. Adedayo Ajidahun 

Academic Editor

PLOS ONE